# Bayesian Model Selection and Misspecification Testing in Imaging Inverse Problems Only from Noisy and Partial Measurements

**Tom Sprunck** [1]  **Marcelo Pereyra** [2]  **Tobías I. Liaudat** [1]

## Abstract

Modern imaging techniques heavily rely on Bayesian statistical models to address difficult image reconstruction and restoration tasks. This paper addresses the objective evaluation of such models in settings where ground truth is unavailable, with a focus on model selection and misspecification diagnosis. Existing unsupervised model evaluation methods are often unsuitable for computational imaging due to their high computational cost and incompatibility with modern image priors defined implicitly via machine learning models. We herein propose a general methodology for unsupervised model selection and misspecification detection in Bayesian imaging sciences, based on a novel combination of Bayesian cross-validation and data fission, a randomized measurement splitting technique. The approach is compatible with any Bayesian imaging sampler, including diffusion and plug-and-play samplers. We demonstrate the methodology through experiments involving various scoring rules and types of model misspecification, where we achieve excellent selection and detection accuracy with a low computational cost.

## 1. Introduction

**Preliminaries**  Modern quantitative and scientific imaging techniques heavily rely on statistical models and inference methods to analyze raw sensor data, reconstruct high-quality images, and extract meaningful information (Bhandari et al., 2022). Despite the diversity of imaging modalities and applications, most statistical imaging methods aim to infer an unknown image $x_\star \in \mathbb{R}^n$ from a measurement

[1]IRFU, CEA, Université Paris-Saclay, F-91191 Gif-sur-Yvette Cedex, France [2]Heriot-Watt University, MACS & Maxwell Institute for Mathematical Sciences, EH14 4AS, Edinburgh, United Kingdom. Correspondence to: Tom Sprunck, Tobías I. Liaudat <tom.sprunck@cea.fr, tobias.liaudat@cea.fr>.

*Proceedings of the 43$^{rd}$ International Conference on Machine Learning*, Seoul, South Korea. PMLR 306, 2026. Copyright 2026 by the author(s).

$y \in \mathbb{R}^m$, modeled as a realization of

$$\mathbf{y} \mid \mathbf{x} = x_\star \sim P(A(x_\star)) \tag{1}$$

where $A$ is an experiment-specific measurement operator representing deterministic physical aspects of the sensing process, and $P$ is a statistical noise model (Kaipio & Somersalo, 2010). Common examples include image denoising, demosaicing, deblurring, and tomographic reconstruction (Bhandari et al., 2022).

A key common feature across statistical imaging is that recovering $x_\star$ from $y$ involves solving an inverse problem that is not well-posed, requiring regularization to stabilize the inversion. The Bayesian paradigm addresses regularization by treating $x_\star$ as a random variable $\mathbf{x}$ and incorporating prior knowledge through the marginal $p(x)$. This prior is then combined with the likelihood function $p(y|x)$ via Bayes' theorem, yielding the posterior distribution

$$p(x|y) = \frac{p(y|x)p(x)}{\int p(y|\tilde{x})p(\tilde{x})\mathrm{d}\tilde{x}},$$

which underpins all inferences about $\mathbf{x}$ having observed $\mathbf{y} = y$ (Robert, 2007). Beyond producing estimators, modern Bayesian imaging methods increasingly quantify uncertainty in the reconstruction, an essential component for reliable interpretation and robust integration with decision-making processes. Of course, modeling choices may strongly influence the delivered inferences, making the development of ever more accurate Bayesian imaging models a continual focus of research.

Modern Bayesian imaging methods increasingly use highly informative image priors encoded by deep learning models that deliver unprecedented estimation accuracy (Heckel, 2025; Mukherjee et al., 2023a). Notable examples of Bayesian imaging frameworks with data-driven priors include plug-and-play Langevin samplers (Laumont et al., 2022a; Renaud et al., 2024; Kemajou Mbakam et al., 2025), denoising diffusion models (Zhu et al., 2023; Daras et al., 2024; Song et al., 2023; MOUFAD et al., 2025), distilled diffusion models (Spagnoletti et al., 2025; Mbakam et al., 2025), flow matching (Martin et al., 2025), and conditional GANs (Bendel et al., 2023; 2024). In addition, while traditional approaches to developing data-driven image priors required large amounts of clean training data,

modern methods increasingly learn image models directly from measurement data (Chen et al., 2023). These models can also be designed to exhibit the mathematical regularity needed for integration into optimization algorithms and Bayesian sampling machinery (Mukherjee et al., 2023b).

However, while already widely deployed in photographic imaging pipelines, leveraging data-driven priors for quantitative and scientific imaging remains challenging due to the stricter requirements for reliability and accuracy. For example, data-driven priors can lead to strongly biased inferences if, during deployment, the encountered image $x_\star$ is poorly represented in the training data. In such cases, highly informative priors may override the likelihood $p(y|x)$, particularly in ill-posed or ill-conditioned problems where the likelihood has poor identifiability. It is therefore essential to equip critical imaging pipelines with the ability to self-diagnose model misspecification. Similarly, multiple data-driven priors and likelihoods may be available for inference, each reflecting different assumptions about the sensing process and the scene; assumptions that are often unverifiable in practice. Hence, robust imaging pipelines must be able to objectively compare alternative models based solely on measurement data.

**Problem statement** We consider the problem of objectively comparing and diagnosing misspecification in Bayesian imaging models, directly from measurement data, without access to ground truth. We focus on modern data-driven image priors encoded by large deep learning models, which are highly informative and may be improper.

**Contributions** We herein propose a statistical methodology for performing Bayesian model selection and misspecification diagnosis in large-scale imaging inverse problems. Our proposed methodology is fully unsupervised, in that the analyses solely use a single noisy measurement $y$. This is achieved by leveraging measurement splitting by noise injection (Pang et al., 2021; Monroy et al., 2025), also known as data fission (Leiner et al., 2025), in order to construct a self-supervising Bayesian cross-validation procedure. The methodology is agnostic to the class of image priors used and fully compatible with modern priors encoded by deep learning models. In addition, the method is computationally efficient and can be straightforwardly integrated within widely used Bayesian imaging sampling strategies, such as Langevin and guided denoising diffusion samplers. We demonstrate the effectiveness of our approach through experiments related to image deblurring with plug-and-play Langevin samplers and denoising diffusion models for photographic and magnetic resonance images, achieving excellent model selection and misspecification detection accuracy even in challenging settings.

## 2. Background

**Prior predictive checking** evaluates the model $p(x|y)$ by comparing the observation $y$ to predictions of $\mathbf{y}$ derived from the model (Gelman et al., 2013). Such checks often use the prior predictive distribution, with density $p(y) = \int p(y|x)p(x)\mathrm{d}x$, or more generally an expected utility loss $\Phi(y) = \int \phi(y,x)p(x)\mathrm{d}x$, where $\phi(y,x)$ quantifies the discrepancy between a possible $x$ and $y$. Prior predictive checks implicitly view $p(\mathbf{x}, \mathbf{y})$ as a generative model for $(\mathbf{x}, \mathbf{y})$ and they provide a useful lens to examine the implications of specific prior and likelihood choices. However, they do not evaluate how well the model supports inference on $\mathbf{x}$ after fitting to $\mathbf{y} = y$, nor do they reveal how specific forms of model misspecification affect particular inferences. Also, prior checks are not well-defined when $p(\mathbf{x})$ is improper, even if the resulting posterior $p(\mathbf{x}|\mathbf{y} = y)$ is well-posed and yields meaningful accurate inferences, as is often the case in Bayesian imaging models.

**Posterior predictive checking** evaluates the model $p(\mathbf{x}, \mathbf{y})$ through the prediction of a new measurement $\mathbf{y}^+ \sim P(Ax_\star)$ stemming from a hypothetical experiment replication, conditionally to $\mathbf{y} = y$ (Gelman et al., 2013). Such checks leverage the posterior predictive distribution, with density $p(y^+|y) = \int p(y^+|x)p(x|y)\mathrm{d}x$ where the unknown image $\mathbf{x}$ is drawn from $p(x|y)$. Posterior predictive checks reveal model misfit by identifying discrepancies between the prediction $\mathbf{y}^+|\mathbf{y} = y$ derived from $p(\mathbf{x}|\mathbf{y} = y)$ and the observed measurement $\mathbf{y} = y$. Again, both application-agnostic and task-specific scoring rules can be used to probe targeted aspects of the model. However, posterior checks are often overly optimistic, as predictions are conditioned on the observed data and thus biased towards agreeing with it (Gelman et al., 2013).

**Bayesian cross validation** is a powerful partial posterior predictive approach that mitigates the bias of conventional posterior predictive checks by holding out part of the data, fitting the model to the remainder, and evaluating predictive performance on the held-out set. This yields more reliable diagnostics, as it breaks the circularity of using the same data for both model fitting and evaluation (Vehtari & Ojanen, 2012; Gelman et al., 2014; Cooper et al., 2024). To make full use of the data, cross-validation employs randomization, repeatedly fitting and evaluating across multiple data partitions. While widely adopted in other domains, Bayesian cross-validation remains largely unexplored in computational imaging, where typically only a single measurement is available. Unfortunately, obtaining two independent measurements of the same scene is often not possible, as imaging experiments occur under conditions that are ephemeral due to dynamic scenes, non-static sensors, and operational constraints.

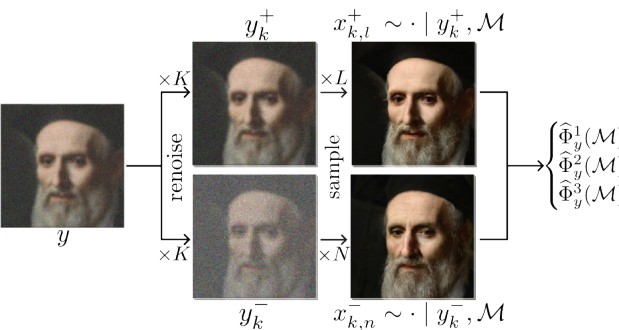

*Figure 1.* Proposed framework to evaluate the model $\mathcal{M}$ on a single measurement $y$. Samples from the posterior distributions relative to the renoised measurements $y^+$ and $y^-$ obtained by measurement splitting are compared via the cross-validation estimators $\widehat{\Phi}_y^i(\mathcal{M})$; averaging over many renoisings ($k$) reduces bias.

**Unsupervised Bayesian model selection** uses strategies similar to model evaluation -namely prior, posterior or partial predictive summaries- but differs fundamentally in its purpose. Model selection aims to rank competing models and identify the one that best explains the observed data, rather than assessing individual model adequacy. Unsupervised Bayesian model selection for computational imaging often relies on the (prior predictive) marginal likelihood $p(y) = \mathbb{E}(p(y|\mathbf{x})) = \int p(y, x)\mathrm{d}x$, particularly through the use of so-called Bayes factors to assess the relative fit-to-data of competing models. However, computing marginal likelihoods for image data is notoriously challenging due to the high dimensionality involved. Early approaches have used harmonic mean estimators (Durmus et al., 2022), while recent efforts have employed nested samplers specifically designed for this task (Skilling, 2006; Cai et al., 2022; McEwen et al., 2023); however, these remain computationally expensive and difficult to scale. Approximations based on empirical Bayesian residuals (Vidal et al., 2021) offer a tractable alternative, but their reliability is limited (Mbakam et al., 2024). One can also consider supervised Bayesian model selection, relying on reference images and controlled experiments. However, this approach is impractical in many application domains where acquiring reliably representative reference data is infeasible.

**Out-of-distribution detection.** In Bayesian imaging, out-of-distribution detection (OOD) methods are predominantly used to identify situations of prior misspecification with respect to datasets. As stated previously, this is especially important when using highly informative priors encoded by large machine learning models. Several supervised OOD methods have recently been proposed in the literature (Liu et al., 2023; Graham et al., 2023; Zhang et al., 2018; Gao et al., 2023). A recent unsupervised method specifically designed for diffusion models was proposed (Shoushtari et al., 2025), but it relies in assuming the availability of a collection of measurements collected with a range of forward operators that collectively spans the entire signal space. To the best of our knowledge, no existing methods can diagnose OOD based on a single measurement or address general Bayesian imaging reconstruction techniques.

## 3. Proposed Method

### 3.1. Bayesian Cross-Validation by Data Fission

We now present our methodology for Bayesian model selection and misspecification testing. Suppose for now the availability of two independent measurements $\mathbf{y}^+, \mathbf{y}^- \sim P(A(x_\star))$ from replication of the experiment. Adopting a partial predictive approach, we evaluate a model $\mathcal{M}$, comprising a prior and likelihood, by computing a summary of the form (Vehtari & Ojanen, 2012)

$$\Psi(\mathcal{M}) = \mathbb{E}_{\mathbf{y}^+, \mathbf{y}^-} \left[ S(p_\mathcal{M}(\mathbf{y}^+|\mathbf{y}^-, y^+)) \right] , \qquad (2)$$
$$= \int S(p_\mathcal{M}(\mathbf{y}^+|\mathbf{y} = y^-), y^+)p_\mathcal{M}(y^-, y^+)\mathrm{d}y^- \, \mathrm{d}y^+ ,$$

where $S : \mathcal{P} \times \mathbb{R}^m \mapsto \mathbb{R}_+$ is a scoring rule (Gneiting & Raftery, 2007) that takes a predictive density $p \in \mathcal{P}$, with $\mathcal{P}$ being a probability measure, and a realization mapping it to a numerical assessment of that prediction. In the case of (2), we summarize the models' capacity to predict $\mathbf{y}^+$ having observed $\mathbf{y}^-$, under the assumptions encoded by $p(y^-, y^+) = \int p(y^+|x)p(y^-|x)p(x)\mathrm{d}x$ as described by model $\mathcal{M}$. With regards to $S$, a classic choice is the logarithmic rule $S(p(\mathbf{y}^+|y^-), y^+) := \log p(y^+|y^-) = \log \int p(y^+|x)p(x|y^-)\mathrm{d}x$, which is known to be strictly proper (Gneiting & Raftery, 2007). Other rules allow probing of $\mathcal{M}$ for particular forms of misspecification; examples tailored for imaging are provided later.

Summaries of the form (2) are usually computed approximately by cross-validation, with K-fold randomization of the data partition. However, implementing Bayesian cross-validation in imaging is challenging, as often only a single data point $y$ is available. To overcome this fundamental difficulty, our approach leverages data fission (Leiner et al., 2025), a form of measurement splitting by noise injection used in computer vision (Pang et al., 2021; Monroy et al., 2025). This leads to a Bayesian cross-validation approach that, from a single measurement $y$, offers a trade-off between accuracy and computational efficiency.

Measurement splitting strategies partition a single observed outcome $\mathbf{y} = y$ from $\mathbf{y} \sim P(A(x_\star))$ into two synthetic measurements $\mathbf{y}^+$ and $\mathbf{y}^-$ that are conditionally independent given $x_\star$. For presentation clarity, we introduce this step for problems involving additive Gaussian noise, and subsequently extend the approach to other noise models. Suppose that $\mathbf{y} \sim \mathcal{N}(A(x_\star), \Sigma)$ and let $\mathbf{w} \sim \mathcal{N}(0, \Sigma)$.

Then, for any $\alpha \in (0, 1)$,

$$
\begin{aligned}
\mathbf{y}^+ &= f_\alpha^-(\mathbf{y}, \boldsymbol{w}) \coloneqq \mathbf{y} + c_\alpha \boldsymbol{w}, \\
\mathbf{y}^- &= f_\alpha^-(\mathbf{y}, \boldsymbol{w}) \coloneqq \mathbf{y} - \boldsymbol{w}/c_\alpha,
\end{aligned}
\tag{3}
$$

with $c_\alpha = \sqrt{\alpha/(1-\alpha)}$ are independent Gaussian variables conditionally to $x_\star$, with mean $A(x_\star)$ and covariance proportional to $\Sigma$. For the specific case of $\alpha = 0.5$, they are i.i.d. with marginal distribution $\mathcal{N}(A(x_\star), 2\Sigma)$. For $\alpha \neq 0.5$, we have that the information in $\mathbf{y}$ is divided unequally between $\mathbf{y}^+$ and $\mathbf{y}^-$; reducing $\alpha$ brings $\mathbf{y}^+$ closer to $\mathbf{y}$ and reduces the correlation between $\mathbf{y}$ and $\mathbf{y}^-$. Equivalent splitting strategies are available for other noise models from the natural exponential family (Monroy et al., 2025), including Poisson and Gamma noise commonly encountered in imaging.

By combining (2) with measurement splitting, our proposed Bayesian cross-validation approach evaluates a model $p_\mathcal{M}(x, y)$ through its capacity to deliver accurate predictions of $f_\alpha^+(y, \mathbf{w})$ from $f_\alpha^-(y, \mathbf{w})$; i.e.,

$$
\Phi_y(\mathcal{M}) = \mathbb{E}_{\mathbf{w}} \left[ \mathbb{E}_{\mathbf{x}|f_\alpha^-(y,\mathbf{w}),\mathcal{M}} \left[ \phi_\mathcal{M}(f_\alpha^+(y, \mathbf{w}), \mathbf{x})) \right] \right] \tag{4}
$$

$$
= \int \phi_\mathcal{M}(f_\alpha^+(y, w), x) \, p_\mathcal{M}(x \mid f_\alpha^-(y, w)) p(w) \mathrm{d}x \mathrm{d}w
$$

where $\phi_\mathcal{M} : \mathbb{R}^m \times \mathbb{R}^n \mapsto \mathbb{R}_+$ quantifies the discrepancy between a possible $x$ and $y^+$, leading to a scoring rule $\mathbb{E}_{\mathbf{x}|f_\alpha^-(y,\mathbf{w}),\mathcal{M}} [\phi_\mathcal{M}(f_\alpha^+(y, \mathbf{w}), \mathbf{x}))]$ for the prediction of $\mathbf{y}^+$ from $y^-$ (related to each other via $\mathbf{x} \sim p_\mathcal{M}(x|y^-)$, which is marginalized out). The expectation over the noise $\mathbf{w}$ plays a role analogous to randomized data partitions in K-fold cross-validation, with $\alpha$ controlling the share of information in $y$ that is held out.

### 3.2. Scoring Rules for Probing Imaging Models

Below, we discuss two specific scoring rules we recommend for imaging applications.

**Likelihood-based rule** To probe the likelihood $p(y|x)$, we use a rule based on the log likelihood $\phi_\mathcal{M}^1(f_\alpha^+(y, \mathbf{w}), \mathbf{x}) = \log p_\mathcal{M}(f_\alpha^+(y, \mathbf{w})|\mathbf{x})$ and obtain

$$
\Phi_y^1(\mathcal{M}) = \mathbb{E}_{\mathbf{w}} \left[ \mathbb{E}_{\mathbf{x}|f_\alpha^-(y,\mathbf{w}),\mathcal{M}} \left[ \log p_\mathcal{M}(f_\alpha^+(y, \mathbf{w})|\mathbf{x}) \right] \right]. \tag{5}
$$

This rule is closely related to the logarithmic score applied to (2) via Jensen's inequality (Jordan et al., 1999). However, we recommend it over the logarithmic score due to its significantly greater numerical stability (Bishop, 2006).

**Posterior-based rule** Consider a severely ill-posed inverse problem where $A$ is severely rank deficient and therefore the observations are not very informative. In that case,

the rule based on the log likelihood will have poor discrimination w.r.t. the properties of the prior. For example, in the case of a linear Gaussian model, $\log p_\mathcal{M}(f_\alpha^+(y, w)|\mathbf{x}) \propto \|f_\alpha^+(y, w) - A\mathbf{x}\|_2^2$ will not be sensitive to information about $p_\mathcal{M}(x|f_\alpha^-(y, \mathbf{w}))$ in the null space of A. In this scenario, we recommend using a rule that incorporates $p_\mathcal{M}(x|f_\alpha^+(y, \mathbf{w}))$, so that there is a direct comparison between $p_\mathcal{M}(x|f_\alpha^+(y, \mathbf{w}))$ and $p_\mathcal{M}(x|f_\alpha^-(y, \mathbf{w}))$ without the action of $A$. For example,

$$
\phi_\mathcal{M}^2(f_\alpha^+(y, \mathbf{w}), \mathbf{x}) = \mathbb{E}_{\mathbf{x}'|f_\alpha^+(y,\mathbf{w}),\mathcal{M}} \left[ s_\rho(\mathbf{x}, \mathbf{x}') \right], \tag{6}
$$

where $s_\rho : \mathbb{R}^k \times \mathbb{R}^k \mapsto \mathbb{R}_+$ is a discrepancy in an embedding space tailored for a particular task, and is generated by the map $\rho : \mathbb{R}^n \mapsto \mathbb{R}^k$. The resulting summary reads

$$
\Phi_y^2(\mathcal{M}) = \mathbb{E}_{\mathbf{w}} \left[ \mathbb{E}_{\mathbf{x}|f_\alpha^-(y,\mathbf{w}),\mathcal{M}} \left[ \mathbb{E}_{\mathbf{x}'|f_\alpha^+(y,\mathbf{w}),\mathcal{M}} \left[ s_\rho(\mathbf{x}, \mathbf{x}') \right] \right] \right]. \tag{7}
$$

A standard choice for the discrepancy would be $s_\rho(x, x') = \|\rho(x) - \rho(x')\|_2$. Depending on the characteristics of the inverse problem and the model, different embedding spaces can be considered. For a distortion-focused comparison, the embedding mapping $\rho(\cdot)$ would be the identity. However, we can use LPIPS-based embedding (Zhang et al., 2018) for a perception-focused comparison, CLIP-based embedding (Radford et al., 2021) for a semantic-focused comparison, or Dinov2 embedding (Oquab et al., 2023) for general-purpose problems.

**Monte Carlo approximation** In practice, we approximate the expectations in the comparison metrics using Monte Carlo sampling. For the likelihood-based metric under Gaussian noise with a diagonal covariance matrix, we compute the negative log-likelihood (omitting the normalization constant) as follows:

$$
\widehat{\Phi}_y^1(\mathcal{M}) = \frac{1}{KN} \sum_{k=1}^K \sum_{n=1}^N \|y + c_\alpha w_k - A(x_{k,n})\|_2^2, \tag{8}
$$

where $x_{k,n}$ follows the posterior $(\mathbf{x}^-|f_\alpha^-(y, w_k), \mathcal{M})$ and $w_k$ is a realization of $\mathcal{N}(0, \sigma I_m)$. For the posterior-based rule with an LPIPS embedding $\rho_\mathrm{L}$, we have

$$
\widehat{\Phi}_y^2(\mathcal{M}) = \frac{1}{KNL} \sum_{k,n,l=1}^{K,N,L} \|\rho_\mathrm{L}(x_{k,n}^-) - \rho_\mathrm{L}(x_{k,l}^+)\|_2, \tag{9}
$$

where $x_{k,n}^-$ and $x_{k,l}^+$ are respectively samples from $(\mathbf{x}^- \mid f_\alpha^-(y, w_k), \mathcal{M})$ and $(\mathbf{x}^+ \mid f_\alpha^+(y, w_k), \mathcal{M})$. Our experiments suggest that the estimators $\widehat{\Phi}_y^1(\mathcal{M})$ and $\widehat{\Phi}_y^2(\mathcal{M})$ are accurate even with few samples. A graphical summary of the method is presented in Fig. 1, and the algorithm is further detailed in Section A, with a brief discussion on the computational cost.

### 3.3. Relation with Posterior Predictive Checks and the Marginal Likelihood

Let us now consider a single splitting noise realization $\mathbf{w} = w$. For ease of presentation we note $y^+ = f_\alpha^+(y, w)$ and $y^- = f_\alpha^-(y, w)$ and use the splitting from (3). If we choose the likelihood $\phi_{\mathcal{M}}^3(y^+, \mathbf{x}) = p_{\mathcal{M}}(y^+|\mathbf{x})$, the proposed metric in Eq. (4) reads

$$\Phi_y^3(\mathcal{M}) = \mathbb{E}_{\mathbf{x}|y^-, \mathcal{M}}\left[p_{\mathcal{M}}(y^+|\mathbf{x})\right] = p_{\mathcal{M}}(y^+|y^-)$$
$$= \int p_{\mathcal{M}}(y^+|x)p_{\mathcal{M}}(x|y^-)\mathrm{d}x \tag{10}$$

which is the posterior predictive check for model $\mathcal{M}$ on the "new" observation $y^+$ conditioned to the previous observation $y^-$. In the limit of $\alpha$ tending to zero, we have that $y^+$ tends to $y$ and $y^-$ to an independent noise realization. Hence, the limiting behavior as $\alpha \to 0^+$ can be made rigorous as follows.

**Proposition 1.** *Assume* $\mathbf{w}$ *admits a continuous, bounded and strictly positive Lebesgue density, and that the density* $y \to p_{\mathcal{M}}(y|x)$ *is continuous and uniformly bounded. Then, for* $p_{\mathbf{w}}$*-a.e. realization* $w$*, with* $y^+, y^-$ *as in* (3),

$$\lim_{\alpha \to 0^+} \Phi_y^3(\mathcal{M}) = \int p_{\mathcal{M}}(y|x)\,p_{\mathcal{M}}(x)\,dx = p_{\mathcal{M}}(y), \tag{11}$$

*i.e., the marginal likelihood of* $y$ *under model* $\mathcal{M}$*.*

The proof is provided in Appendix C.3.

The main difference between the two previous formulations is that, in the first one, $\mathbf{x}$ follows the partial posterior $p(x|y^-, \mathcal{M})$ instead of the prior $p(x|\mathcal{M})$. Conditioning on the variable $y^-$, a noisier version of $y$, greatly helps to improve the behavior of the estimator. Each sample from the pseudo posterior is more likely to have a higher likelihood value and to contribute to the calculation of the expectation. We approximate the estimator (11) as

$$\widehat{p}_{\mathcal{M}}(y^+|y^-) = \frac{1}{KN}\sum_{k=1}^K\sum_{n=1}^N p_{\mathcal{M}}(y + c_\alpha w_k|x_{k,n}), \tag{12}$$

where $x_{k,n}$ follows the posterior $\mathbf{x} \mid y - w_k/c_\alpha, \mathcal{M}$ and $w_k$ is a realization of $\mathcal{N}(0, \sigma^2 I_m)$.

The role of $\alpha$ is to control the split of information between the conditioning variable $y^-$, easing the evidence calculation, and the estimator variable $y^+$, which we use to compute the marginal likelihood and evaluate model fit-to-data.

## 4. Experimental Results

### 4.1. Error Analysis in the Gaussian Case

We first study a toy Gaussian model, designed to illustrate the proposed methodology under various degrees of

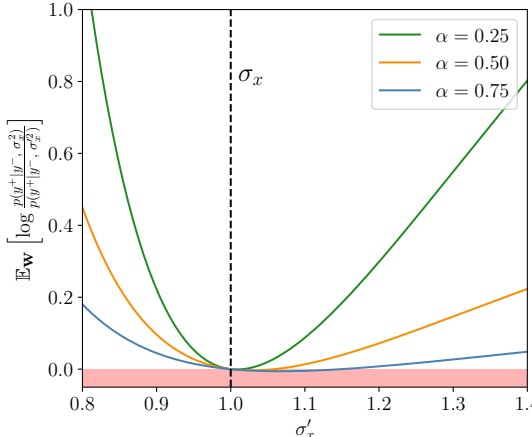

*Figure 2.* Log difference between $p(y^+|y^-, \sigma_x^2)$ and $p(y^+|y^-, \sigma_x'^2)$ as a function of $\sigma_x'$ and for different $\alpha$, averaged over the injected noise $\mathbf{w}$. The true prior standard deviation is $\sigma_x = 1$.

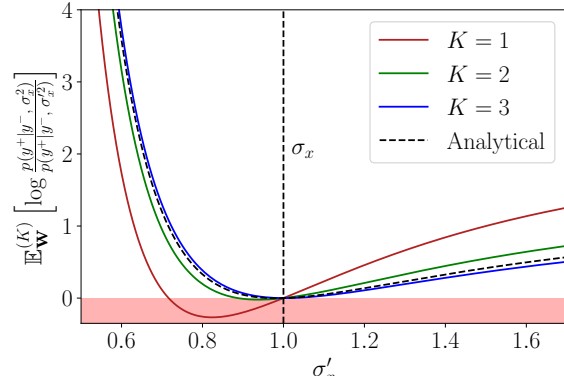

*Figure 3.* Log difference between $p(y^+|y^-, \sigma_x^2)$ and $p(y^+|y^-, \sigma_x'^2)$ as a function of $\sigma_x'$ and for different numbers of noise realizations $K$, with $\alpha = 0.5$. The limit value when $K \to \infty$ is represented as a dotted line.

model misspecification, model size, and splitting parameter $\alpha$. We assume that $\mathbf{y} = \mathbf{x} + \mathbf{e}$, where $\mathbf{e} \sim \mathcal{N}(0, \sigma^2 I_m)$ and $\mathbf{x} \sim \mathcal{N}(0, \sigma_x^2 I_m)$ are independent of $\mathbf{e}$. For ease of presentation, we use the notation $y^+$ and $y^-$ from Section 3.3 . For this model, we have a Gaussian posterior $p(x|y^-) = p_{\mathcal{N}}(x|\frac{\alpha}{\alpha\sigma_x^2 + \sigma^2}y^-, \frac{\sigma^2\sigma_x^2}{\sigma^2 + \alpha\sigma_x^2}I_m)$, where the predictive density $p(y^+|y^-)$ is tractable (see Section C of the supplementary material (SM)).

We draw realizations from $\mathbf{y}$ with $\sigma_x^2 = 1, \sigma^2 = 0.05$, and posit that $\mathbf{x} \sim \mathcal{N}(0, \sigma_x'^2 I_m)$ to study the impact of misspecification. Fig. 2 shows the expectation of the marginal log-likelihood ratio $\log(p(\mathbf{y}^+|\mathbf{y}^-, \sigma_x^2)/p(\mathbf{y}^+|\mathbf{y}^-, \sigma_x'^2))$ as a function of $\sigma_x'$ for different values of $\alpha$, as estimated by averaging over $K = 250$ realizations of $\mathbf{w}$ and when $m = 1000$. We observe that, as expected, model discrimination improves as $\alpha$ decreases and more information is held out in $\mathbf{y}^+$ for model evaluation (recall that $\alpha \to 0$ leads to the marginal likelihood, which is excellent for model discrim-

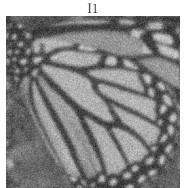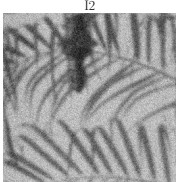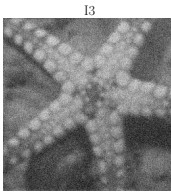

*Figure 4.* Examples of blurred measurements, generated by using the blur kernel $\kappa_{\mathcal{G}}(2)$.

| | Single Shot | Few Shot |
|---|---|---|
| Ours (w. $\widehat{\Phi}^1$) | 86.7% | 100% |
| Bayes Res. (Vidal et al., 2021) | 40.0% | 40.0% |
| EB Res. (Vidal et al., 2021) | 46.7% | 60.0% |

*Table 1.* Accuracy of likelihood model selection, using the the proposed summary $\widehat{\Phi}^1$ and two variants of the baseline method (Vidal et al., 2021), from a single measurement (single shot) or three measurements (few shot).

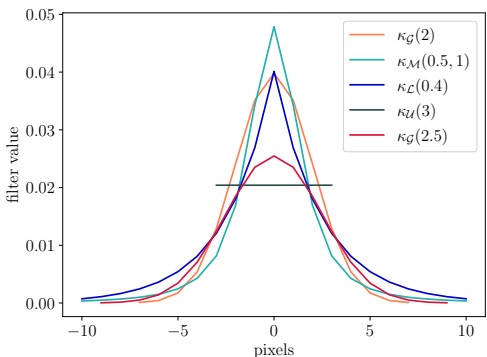

*Figure 5.* Profile of the considered blur kernels, their similarity makes model selection difficult.

ination but often computationally intractable). Moreover, we see in Fig. 3 that averaging $K$ realizations of $\mathbf{w}$ reduces the bias introduced by measurement splitting, akin to randomization in K-fold cross-validation. With regard to computational cost, reducing $\alpha$ increases the number of Monte Carlo samples required to reliably approximate $p(y^+|y^-)$, highlighting a trade-off between evaluation accuracy and efficiency (see SM, Section C).

### 4.2. Unsupervised Likelihood Model Selection

We now consider an image deblurring problem $\mathbf{y} = Ax_\star + \mathbf{e}$, where $\mathbf{e} \sim \mathcal{N}(0, \sigma^2 I_m)$ with $\sigma = 0.1$ and where $A$ is a circulant blur operator implementing the action of a blurring kernel $\kappa_{\mathrm{GT}}$. Given blur kernels from the Moffat, Laplace, Uniform, and Gaussian parameter families presented in Fig. 5, set to be as close as possible, we wish to identify the ground truth kernel relating a measurement $y$ to $x_\star$. See SM, Section D, for the parametric kernel forms.

For each test image in Fig. 4, of size $256 \times 256$ pixels, we generate 5 noisy measurements using the 5 kernels as ground truth. We then compute the value of the log-likelihood-based estimator $\widehat{\Phi}^1$ for each observation and each of the considered blur kernels, seeking to use $\widehat{\Phi}^1$ to identify the correct kernel. We adopt a Langevin PnP approach (Laumont et al., 2022b) and use the gradient step denoiser (Hurault et al., 2021) as prior together with the SK-ROCK algorithm (Abdulle et al., 2018) for posterior sampling. To compute $\widehat{\Phi}^1_y$, we set $\alpha = 0.5$ and draw $K = 10$ realizations of $\mathbf{w}$ and $N = 100$ posterior samples per realization. Tab. 1 reports the model selection accuracy for our method when each observation is analyzed separately (single shot), and when we assume that the blur kernel is shared across the three images (few shot). We observe that our method correctly identifies the blur kernel from a single measurement in over 85% of the cases, and with perfect accuracy when pooling three measurements. For comparison, we also report the Bayesian residual method (Vidal et al., 2021) and the empirical Bayesian variant that improves model selection performance by automatically calibrating model parameters (Vidal et al., 2020). Their accuracy is noticeably lower, in the order of 40% to 60%. Please see SM, Section D for implementation details.

### 4.3. Prior Selection and OOD Detection

We now explore our estimator's ability to objectively compare different image priors and diagnose prior misspecification in OOD situations. We focus on priors represented by denoising diffusion models and use the DiffPIR algorithm (Zhu et al., 2023) for posterior sampling.

#### 4.3.1. DEBLURRING OF NATURAL IMAGES

We first consider a deblurring problem in natural images of size $256 \times 256$ pixels. We use two Diffusion UNet models from Choi et al. (Choi et al., 2021) as priors, which were trained on color images of FFHQ and AFHQ-dogs, respectively. We define the forward operator $A$ as an isotropic Gaussian blurring operator with bandwidth $\sigma_\kappa \in \{0.5, 2, 5\}$ to reflect mild, moderate and high blur, and set the Gaussian noise level to $\sigma = 0.05$. Two datasets are defined: a reference in distribution (ID) subset of 60 images from FFHQ (Karras et al., 2019), and test

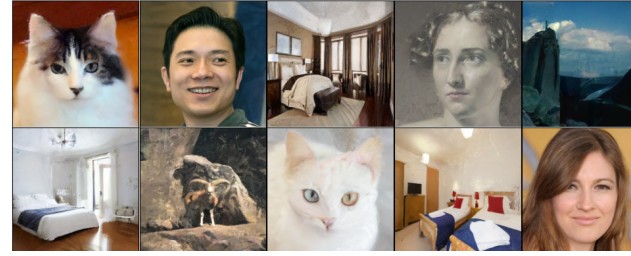

*Figure 6.* Posterior samples from $p(x|y^-)$ for $\alpha = 0.1, \sigma_\kappa = 0.5$, for some test natural image examples.

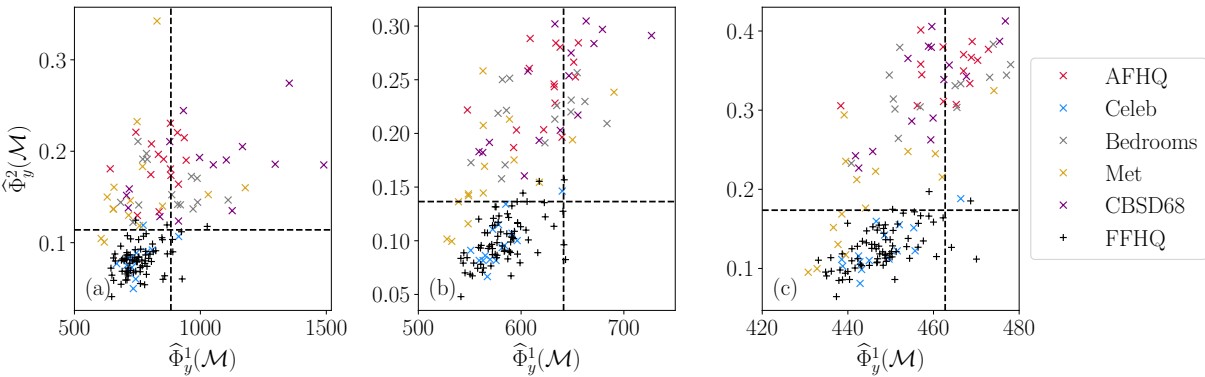

*Figure 7.* OOD detection on natural images, respectively for (a) $\sigma_\kappa = 0.5$, (b) $\sigma_\kappa = 2$ and (c) $\sigma_\kappa = 5$. The dotted lines indicates the testing threshold (95-th percentile of the test statistic over the reference FFHQ subset).

|  |  | $\sigma_\kappa = 0.5$ | $\sigma_\kappa = 2$ | $\sigma_\kappa = 5$ |
|---|---|---|---|---|
| Type I Error | FFHQ | 0% | 6.7% | 0% |
|  | Celeb | 6.7% | 6.7% | 6.7% |
| Power | Moderate OOD | 87% | 73% | 60% |
|  | Strong OOD | 100% | 100% | 100% |

*Table 2.* Type I error rate (incorrect rejection of ID samples from FFHQ, Celeb) and Power (correct rejection of moderate OOD (Met) and strong OOD (bedrooms, CBSD68, AFHQ) examples).

dataset composed of 90 images from AFHQ (Choi et al., 2020), CelebA-HQ (Karras et al., 2018), LSUN-Bedrooms (Yu et al., 2015), Met-Faces (Karras et al., 2020), CBSD68 (Martin et al., 2001), and FFHQ, representing different degrees of prior misspecification. Indeed, while Bedrooms, CBSD68 and AFHQ images are strongly OOD, the images from Met are only moderately OOD and constitute a limit case. Celeb images stem from a different dataset but should be considered ID. Some of the blurred observations are represented in SM, Fig. 22.

We compute the estimators $\widehat{\Phi}_y^1$ and $\widehat{\Phi}_y^2$ for the reference images and test images, using $K = 10$ noise realizations with $N = 20$ samples each, and $\alpha = 0.1$. Fig. 7 depicts the values of the estimators $\widehat{\Phi}_y^1$ and $\widehat{\Phi}_y^2$ on the reference dataset and test datasets. Note that $\widehat{\Phi}_y^2$ is highly sensitive to OOD situations, while $\widehat{\Phi}_y^1$ has more limited value in this case.

For OOD detection, we consider the null hypothesis to be "in distribution", and we define a simple statistical test by setting a threshold at the 95-th percentile of $\widehat{\Phi}_y^2$ over the reference dataset. Tab. 2 reports the Type I error probability and power for each test subset, at significance level 5%. Observe that testing with $\widehat{\Phi}_y^2$ achieves a Type I error close to the desired 5% on the two ID datasets, and excellent power on the moderate and strongly OOD cases. As expected, the power of the test decreases as the blur strength $\sigma_\kappa$ increases and removes fine detail, especially in mild OOD cases.

The effectiveness of $\widehat{\Phi}_y^2$ stems from the fact that, when $x_\star$ is

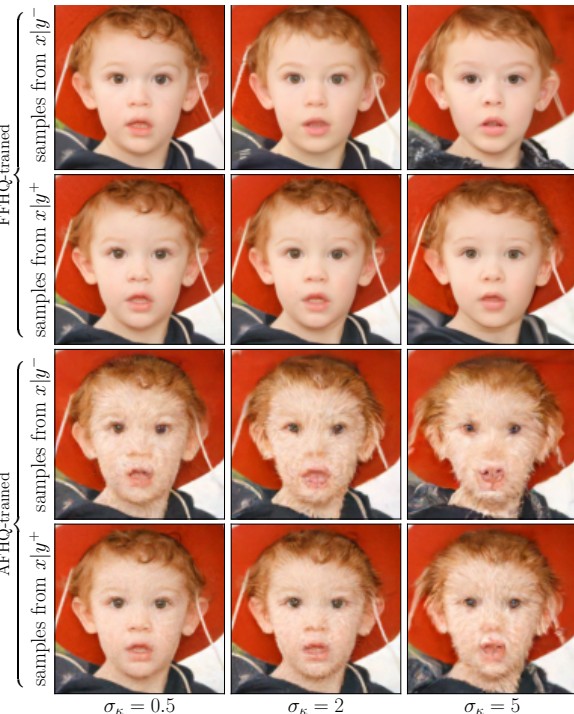

*Figure 8.* Samples from $x|y^-$ and $x|y^+$ for a correctly specified model (FFHQ) and a misspecified model (AFHQ), where $y$ is obtained by degrading an FFHQ image with increasing blur ($\sigma_\kappa = 0.5$, 2, 5).

OOD and $\alpha$ is small, the noise imbalance between $y^+$ and $y^-$ creates a noticeable perceptual discrepancy between the posterior samples from $p(x|y^+)$ and $p(x|y^-)$. To illustrate this, Figure 8 depicts samples from the posterior distributions $p(x|y^+)$ and $p(x|y^-)$ under both well-specified and strongly misspecified priors. As the blur strength increases, perceptual hallucinations become more pronounced in the OOD model's reconstructions. This effect persists, though more weakly, under mildly misspecified priors, resulting in a drop in detection power at high blur levels. To illustrate a mild OOD situation, Figure 9 shows a Met-Faces exam-

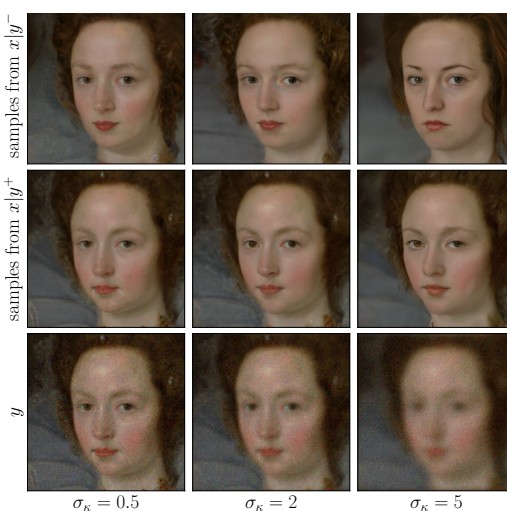

Figure 9. Measurements $y$ and samples from $x|y^-$ and $x|y^+$ for the FFHQ-trained model, where $y$ is obtained by blurring a Met-Faces image.

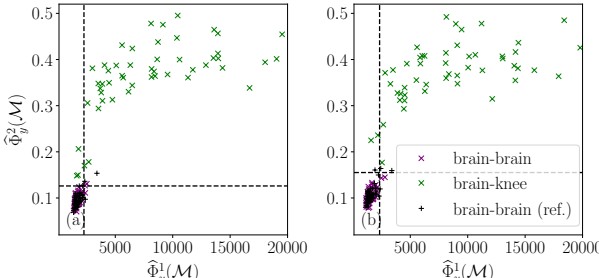

Figure 10. OOD detection on MRI scans at the acceleration factors (a) $R = 4$ and (b) $R = 8$. The dotted lines indicates the testing threshold, which corresponds to the 95-th percentile of $\widehat{\Phi}_y^2$ (respectively $\widehat{\Phi}_y^1$) on the reference brain scan subset.

ple that is correctly identified as OOD for $\sigma_\kappa = 0.5$ and $\sigma_\kappa = 2$, but misclassified for $\sigma_\kappa = 5$.

### 4.3.2. MRI RECONSTRUCTION

We now consider a single-coil MRI image reconstruction problem (see SM, Section E.6). We use two diffusion priors from Shoushtari et al. (2025), which are trained on brain and knee images from the FastMRI dataset respectively (Zbontar et al., 2018; Knoll et al., 2020). We consider the brain dataset as ID. We proceed similarly to the previous experiment and extract brain and knee scans from FastMRI to compose the ID and OOD datasets. For this experiment, we slightly increase $\alpha$ to $0.25$ to reduce the noise injected to $\mathbf{y}^-$, which allows reducing the number of noise realizations to $4$ and the number of steps to $10$. We define a reference dataset of $50$ brain scans to compute the 95-th percentile of $\widehat{\Phi}_y^2$, and compose a test set of $50$ ID and $50$ OOD images. We set the measurement noise to $0.1$ in all experiments and consider an acceleration factor $R$ of $\times 4$ or $\times 8$ for the forward operator; increasing $R$ makes the estimation problem more challenging.

The values of the estimators $\widehat{\Phi}_y^1$ and $\widehat{\Phi}_y^2$ and for the brain-trained model are represented in Fig. 10 for $R = 4$ and the more challenging case $R = 8$. In both cases, we observe an excellent discrimination between ID and OOD data points for both estimators. This result can be explained by the fact that the brain images comprise few learnable features that can be transposed to knee images. The brain model mainly learns the complex structures (gyri) present on the surface of the brain, which are completely absent from knee scans, and often hallucinates these structures in knee images.

Moreover, to evaluate OOD detection accuracy, Tab. 3 re-

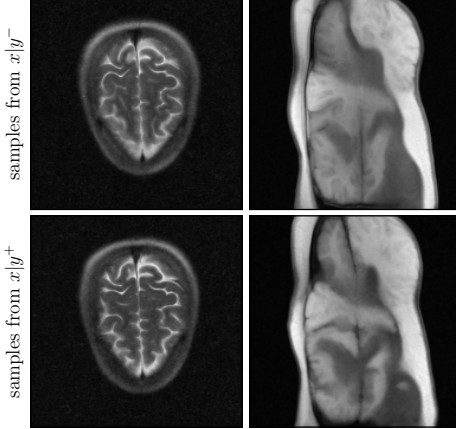

Figure 11. Samples from $x|y^-$ and $x|y^+$ for the brain-trained diffusion model on ID and OOD examples.

|  | $R = 4$ | | $R = 8$ | |
|---|---|---|---|---|
|  | $\widehat{\Phi}_y^2$ | $\widehat{\Phi}_y^1$ | $\widehat{\Phi}_y^2$ | $\widehat{\Phi}_y^1$ |
| Type I Error | 4% | 6% | 0% | 4% |
| Power | 100% | 94% | 100 % | 96% |

Table 3. Type I error rate (incorrect rejection of brain examples), Power (correct rejection of knee examples).

ports the Type I error probability and testing power obtained with each estimator; we observe that they both achieve excellent performance. For completeness, we also report the results for single-shot model selection against a model trained on knee scans in SM, Section E.6. Lastly, Fig. 11 shows examples of samples from $p(x|y^-)$ and $p(x|y^+)$ for ID and OOD cases. Once again, we observe that the OOD case exhibits substantial variability in perceptual details, largely hallucinated by the prior.

### 4.4. Additional Experiments and Ablation Studies

In Section B, we benchmark the proposed method against a state-of-the-art proximal nested sampling algorithm (McEwen et al., 2023), which allows us to compare our

approach against an estimation of the Bayesian evidence. We study the impact of $\alpha$ and the convergence speed of the estimators in Section E.1. In Section E.2, we further present a model selection experiment for the same deblurring problem introduced in Section 4.3.1. We then provide an ablation study using different state-of-the-art embeddings in Section E.3, and we highlight the method's robustness to sample quality degradation in Section E.5. Finally, in Section E.4, we illustrate the adaptability of the proposed method by repeating the same experiment under Poisson noise.

## 5. Discussion and Conclusions

We introduced a Bayesian cross-validation framework for unsupervised model selection and misspecification testing in imaging inverse problems, with a focus on the objective comparison of likelihood functions and data-driven priors encoded by large-scale machine learning models. Leveraging data fission, the proposed method operates using only a single measurement, which is partitioned into two noisier measurements according to a parameter $\alpha$ that governs the amount of information reserved for model evaluation, as well as the trade-off between evaluation accuracy and computational cost. As the marginal likelihood, a gold standard for Bayesian model selection, is recovered in the limit as $\alpha \to 0$ and a specific choice of scoring rule, our approach can be viewed as a relaxation that sacrifices some accuracy for significant gains in efficiency. We propose two main scoring rules for evaluating Bayesian imaging models: a likelihood-based rule, well-suited for assessing likelihood functions, and a perceptual posterior-based rule, which effectively evaluates priors. Furthermore, we demonstrate the effectiveness of the proposed approach through a series of numerical experiments on image photographic deblurring and MRI reconstruction, showcasing its ability to compare likelihoods and image priors, as well as accurately diagnose prior misspecification in both mild and strong out-of-distribution settings.

### ACKNOWLEDGEMENTS

This work was granted access to the HPC resources of IDRIS under the allocation 2024-AD011015754 and 2025-AD011015754R1 made by GENCI. MP acknowledges support by UKRI Engineering and Physical Sciences Research Council (EP/Z534481/1).

### Software and Data

The code to reproduce the experiments is available at https://github.com/aleph-group/Priors_selection, and the models and ground truth images can be downloaded from https://zenodo.org/records/17484892.

## Impact Statement

This paper presents work whose goal is to advance the field of Machine Learning. There are many potential societal consequences of our work, none which we feel must be specifically highlighted here.

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

# A. Implementation Details

## A.1. Computing Estimators $\widehat{\Phi}_y^1(\mathcal{M}), \widehat{\Phi}_y^2(\mathcal{M})$

---

**Algorithm 1** Pseudo-code for computing the estimators $\widehat{\Phi}_y^1(\mathcal{M})$ and $\widehat{\Phi}_y^2(\mathcal{M})$.

---

**Input** : Measurements vector $y$, $\alpha \in ]0,1[$
**Output:** Estimators $\widehat{\Phi}_y^1(\mathcal{M}), \widehat{\Phi}_y^2(\mathcal{M})$
$(\widehat{\Phi}_y^1(\mathcal{M}), \widehat{\Phi}_y^2(\mathcal{M})) \longleftarrow (0,0)$
**for** $k \longleftarrow 1$ **to** $K$ **do**
    Draw $w_k \sim P$
    $y_k^- \leftarrow f_\alpha^-(y, w_k)$
    $y_k^+ \leftarrow f_\alpha^+(y, w_k)$
    **for** $n \longleftarrow 1$ **to** $N$ **do**
        Draw $x_{k,n}^- \sim x \mid y_k^-$
        $\widehat{\Phi}_y^1(\mathcal{M}) \leftarrow \widehat{\Phi}_y^1(\mathcal{M}) + \log p_{\mathcal{M}}(y_k^+ | x_{k,n}^-)$
        **for** $l \longleftarrow 1$ **to** $L$ **do**
            Draw $x_{k,l}^+ \sim x \mid y_k^+$
            $\widehat{\Phi}_y^2(\mathcal{M}) \leftarrow \widehat{\Phi}_y^2(\mathcal{M}) + s_\rho(x_{k,n}^-, x_{k,l}^+)$.

$\widehat{\Phi}_y^1(\mathcal{M}) \leftarrow \frac{1}{KN}\widehat{\Phi}_y^1(\mathcal{M})$
$\widehat{\Phi}_y^2(\mathcal{M}) \leftarrow \frac{1}{KLN}\widehat{\Phi}_y^2(\mathcal{M})$

---

Algorithm 1 presents pseudo-code to compute the quantities $\Phi_y^1(\mathcal{M})$ and $\Phi_y^2(\mathcal{M})$ via Monte-Carlo integration, using $K$ instances of noise splitting, and $N \times L$ posterior samples per split.

In practice, the computational cost of evaluating the likelihood and the scoring rules is negligible compared to generating posterior samples. Since each split is independent, it is therefore more efficient to generate all posterior samples from $x \mid y_k^+$ and $x \mid y_k^-$ in parallel and subsequently aggregate the resulting scores.

## A.2. Computational Overhead

We provide in this section some insights into the computational costs involved in our method. While the cost of the proposed method is higher than the cost of computing a point estimator, it remains significantly lower than existing unsupervised methods for model selection. Indeed, the proposed estimators depend on global statistics and hence converge quickly, producing reliable estimates even for small values of $N$, as shown in Fig. 21. Comparatively, computing the Bayesian model evidence via thermodynamic integration or nested sampling requires orders of magnitude more posterior samples and often struggles in high-dimensional inference. We report in Tab. 4 the run times for the experimental setup of Section 4.3.1, *i.e.*, $K = 10$ and $N = 20$, using two different samplers. We draw here a single sample from the posterior distribution $x|y^+$ for each split ($L = 1$), resulting in 210 total posterior samples. We do not include the time necessary to aggregate the likelihoods or LPIPS loss, as it is negligible when compared to the sampling time. These results were obtained using a single 16 GB Nvidia V100. Note that these run times could be greatly improved by making better use of GPU resources (*e.g.*, we only used a batch size of for the DiffPIR) and by leveraging modern distilled samplers that produce excellent samples in as little as 4-10 NFEs.

| Sampler | Total time (s) | Average time per sample (s) |
|---|---|---|
| PNP (GSD prior) | 323 | 1.54 |
| DiffPIR | 1802 | 8.58 |

*Table 4.* Sampling times for a single image for the experiments of Section 4.3.1.

## B. Comparison to a Proximal Nested Sampling Algorithm

We validate the proposed method by comparing its estimators with a state-of-the-art proximal-based nested sampling algorithm, ProxNest (Cai et al., 2022), which estimates the Bayesian model evidence. This method was extended to deep learning priors in McEwen et al. (2023). We adopt the denoising experimental setting of McEwen et al. (2023), which compares a range of image priors for denoising galaxy images. We extend the original study, which considered a conventional wavelet sparsity prior and a pretrained DnCNN denoiser, by additionally including a more expressive DRUNet denoiser.

For each of these three models, we compute the log evidence (logZ) using the ProxNest algorithm, the two statistics $\Phi_u^1(\mathcal{M})$ and $\Phi_y^2(\mathcal{M})$ using our proposed method, and two supervised model comparison metrics: (i) the PSNR of the posterior mean of $x \mid y$ (approximated by reusing the generated samples for the $\Phi$ statistics), and (ii) the PSNR of the maximum-a-posteriori (MAP) estimate computed with an appropriate optimization algorithm. These supervised metrics are reported solely for validation, since in realistic applications one would not have access to the ground-truth image $x$.

| Prior ($\mathcal{M}$) | $\Phi_y^1(\mathcal{M}) \downarrow$ | $\Phi_y^2(\mathcal{M}) \downarrow$ | PSNR $\mathbb{E}[x|y^-] \uparrow$ (dB) | MAP PSNR $\uparrow$ (dB) | logZ $\uparrow$ |
|---|---|---|---|---|---|
| WAV db6 | 83.38 | 0.0648 | 21.25 | 31.48 | -1156.63 |
| DnCNN | 30.21 | 0.0372 | 29.86 | 31.59 | -503.84 |
| DRUNet | **22.23** | **0.0223** | **32.86** | **34.11** | **-312.66** |

*Table 5.* Unsupervised estimators $\Phi_y^1(\mathcal{M}), \Phi_y^2(\mathcal{M})$, logZ (log Bayesian evidence or log marginal likelihood) and supervised PSNR between the ground truth $x$ and MAP and posterior mean reconstructions for the denoising problem on galaxy images. The logZ was computed using the ProxNest algorithm (McEwen et al., 2023).

Table 5 presents the numerical results. We observe that the model rankings obtained from both $\Phi_1$ and $\Phi_2$ match those of the Bayesian log evidence computed by the ProxNest algorithm. The results are also in agreement with the performance of each model, as reported by the PSNR.

## C. Analysis in the Gaussian Case

### C.1. Derivation of the Analytical Formulas

We compute here the formula for $p(y^+|y^-)$ for $y = x + e$, where $x \sim \mathcal{N}(0, \sigma_x^2 I_m)$ and $e \sim \mathcal{N}(0, \sigma^2 I_m)$. Recall that $y^+ = y + \sqrt{\frac{\alpha}{1-\alpha}}w$ and $y^- = y - \sqrt{\frac{1-\alpha}{\alpha}}w$, with $w \sim \mathcal{N}(0, \sigma^2 I_m)$.

We have:

$$p(y^+|y^-) = \mathbb{E}_{x'|y^-}\big[p(y^+|x')\big] = \int p(y^+|x')\frac{p(y^-|x')p(x')}{p(y^-)}dx'. \tag{13}$$

We can thus write:

$$p(y^+|y^-) = \int \frac{\alpha^{m/2}e^{-\frac{\alpha}{2\sigma^2}\|x'-y^-\|^2}}{(2\pi)^{m/2}\sigma^m} \frac{(1-\alpha)^{m/2}e^{-\frac{1-\alpha}{2\sigma^2}\|x'-y^+\|^2}}{(2\pi)^{m/2}\sigma^m} \frac{e^{-\frac{1}{2\sigma_x^2}\|x'\|^2}}{(2\pi)^{m/2}\sigma_x^m} \frac{(2\pi)^{m/2}(\alpha\sigma_x^2 + \sigma^2)^{m/2}}{\alpha^{m/2}} e^{\frac{\alpha}{2(\alpha\sigma_x^2+\sigma^2)}\|y^-\|^2} dx' \tag{14}$$

$$= \int \frac{\big[(1-\alpha)(\alpha\sigma_x^2 + \sigma^2)\big]^{m/2}}{(2\pi)^m \sigma^{2m}\sigma_x^m} e^{-\frac{\alpha}{2\sigma^2}\|x'-y^-\|^2 - \frac{1-\alpha}{2\sigma^2}\|x'-y^+\|^2 - \frac{1}{2\sigma_x^2}\|x'\|^2 + \frac{\alpha}{2(\alpha\sigma_x^2+\sigma^2)}\|y^-\|^2} dx'. \tag{15}$$

The first part of the exponent can be factorized as:

$$-\frac{1-\alpha}{2\sigma^2}\|x'-y^+\|^2 - \frac{\alpha}{2\sigma^2}\|x'-y^-\|^2 - \frac{1}{2\sigma_x^2}\|x'\|^2 = -\frac{\sigma^2 + \sigma_x^2}{2\sigma^2\sigma_x^2}\|x'\|^2 + \frac{1}{\sigma^2}x' \cdot y - \frac{1-\alpha}{2\sigma^2}\|y^+\|^2 - \frac{\alpha}{2\sigma^2}\|y^-\|^2 \tag{16}$$

$$= -\frac{\sigma^2 + \sigma_x^2}{2\sigma^2\sigma_x^2}\|x' - \frac{\sigma_x^2}{\sigma^2 + \sigma_x^2}y\|^2 + \frac{\sigma_x^2}{2\sigma^2(\sigma^2 + \sigma_x^2)}\|y\|^2 \tag{17}$$

$$- \frac{1-\alpha}{2\sigma^2}\|y^+\|^2 - \frac{\alpha}{2\sigma^2}\|y^-\|^2.$$

Integrating over $x'$ yields:

$$p(y^+|y^-) = \frac{((1-\alpha)(\alpha\sigma_x^2 + \sigma^2))^{m/2}}{(2\pi)^{m/2}\sigma^m(\sigma^2 + \sigma_x^2)^{m/2}} e^{\frac{\sigma_x^2}{2\sigma^2(\sigma^2+\sigma_x^2)}\|y\|^2 - \frac{\alpha^2\sigma_x^2}{2\sigma^2(\alpha\sigma_x^2+\sigma^2)}\|y^-\|^2 - \frac{1-\alpha}{2\sigma^2}\|y^+\|^2}. \tag{18}$$

Finally, expanding the norms in the exponential and refactoring leads to:

$$p(y^+|y^-) = \frac{((1-\alpha)(\alpha\sigma_x^2 + \sigma^2))^{m/2}}{(2\pi)^{m/2}\sigma^m(\sigma^2 + \sigma_x^2)^{m/2}} e^{-\frac{1}{2(\alpha\sigma_x^2+\sigma^2)}\left\|\sqrt{\frac{1-\alpha}{\sigma^2+\sigma_x^2}}\sigma y + \frac{\sqrt{\alpha(\sigma^2+\sigma_x^2)}}{\sigma}w\right\|^2}. \tag{19}$$

Note that as $\alpha \to 0$, we recover the density of $\mathbf{y}$, which is given by: $p(y) = \frac{1}{(2\pi)^{m/2}(\sigma_x^2+\sigma^2)^{m/2}} e^{-\frac{1}{2(\sigma_x^2+\sigma^2)}\|y\|^2}$. The value vanishes to zero as $\alpha \to 1$ and the noise in $y^+$ is amplified. Assuming a misspecified deviation $\sigma_x'$, the log Bayes Factor (BF) is as follows,

$$BF = \log\frac{p(y \mid \sigma_x)}{p(y \mid \sigma_x')} = \frac{m}{2}\log\frac{\sigma^2 + \sigma_x'^2}{\sigma^2 + \sigma_x^2} + \frac{1}{2}\|y\|^2\left(\frac{1}{\sigma^2 + \sigma_x'^2} - \frac{1}{\sigma^2 + \sigma_x^2}\right) \tag{20}$$

and its counterpart is:

$$\log\frac{p(y^+ \mid y^-, \sigma_x)}{p(y^+ \mid y^-, \sigma_x')} = \frac{m}{2}\log\frac{(\sigma^2 + \alpha\sigma_x^2)(\sigma^2 + \sigma_x'^2)}{(\sigma^2 + \alpha\sigma_x'^2)(\sigma^2 + \sigma_x^2)} \tag{21}$$

$$+ \frac{1}{2}\left(\frac{\left\|\sqrt{\frac{1-\alpha}{\sigma^2+\sigma_x'^2}}\sigma y + \frac{\sqrt{\alpha(\sigma^2+\sigma_x'^2)}}{\sigma}w\right\|^2}{\sigma^2 + \alpha\sigma_x'^2} - \frac{\left\|\sqrt{\frac{1-\alpha}{\sigma^2+\sigma_x^2}}\sigma y + \frac{\sqrt{\alpha(\sigma^2+\sigma_x^2)}}{\sigma}w\right\|^2}{\sigma^2 + \alpha\sigma_x^2}\right) \tag{22}$$

We can also compute the analytical value of $\mathbb{E}_{\mathbf{w}}[\log p(y^+|y^-)]$, *i.e* the average over splits as depicted in Fig. 3. Let us write

$$p(y^+|y^-) = h_1 e^{-\frac{1}{2}\|h_2 y + h_3 w\|^2}, \text{ where: } \begin{cases} h_1 = \frac{((1-\alpha)(\alpha\sigma_x^2+\sigma^2))^{m/2}}{(2\pi)^{m/2}\sigma^m(\sigma^2+\sigma_x^2)^{m/2}} \\ h_2 = \sigma\sqrt{\frac{1-\alpha}{(\sigma^2+\sigma_x^2)(\alpha\sigma_x^2+\sigma^2)}} \\ h_3 = \frac{1}{\sigma}\sqrt{\frac{\alpha(\sigma^2+\sigma_x^2)}{\alpha\sigma_x^2+\sigma^2}}. \end{cases} \tag{23}$$

Recall that $\mathbf{w}$ and $\mathbf{y}$ are independent; averaging over $\mathbf{w}$ cancels out the dot product $y^T w$ and we get:

$$\mathbb{E}_{\mathbf{w}}[\log p(y^+|y^-)] = \log h_1 - \frac{1}{2}\left(h_2^2\|y\|^2 + 2h_2 h_3 y^T \mathbb{E}_{\mathbf{w}}(w) + h_3^2\mathbb{E}_{\mathbf{w}}(\|w\|^2)\right) = \log h_1 - \frac{1}{2}\left(h_2^2\|y\|^2 + h_3^2 m\sigma^2\right). \tag{24}$$

Finally, we have:

$$\mathbb{E}_{\mathbf{w}}[\log p(y^+|y^-)] = \frac{m}{2}\log\frac{(1-\alpha)(\alpha\sigma_x^2 + \sigma^2)}{2\pi\sigma^2(\sigma^2 + \sigma_x^2)} - \frac{1}{2}\left(\frac{\sigma^2(1-\alpha)\|y\|^2}{(\sigma^2 + \sigma_x^2)(\alpha\sigma_x^2 + \sigma^2)} + \frac{m\alpha(\sigma^2 + \sigma_x^2)}{\alpha\sigma_x^2 + \sigma^2}\right), \tag{25}$$

and the expected log ratio with respect to splitting noise for a misspecified deviation is:

$$\mathbb{E}_{\mathbf{w}}\left[\log\frac{p(y^+ \mid y^-, \sigma_x)}{p(y^+ \mid y^-, \sigma_x')}\right] = \frac{m}{2}\left(\log\frac{(\sigma^2 + \alpha\sigma_x^2)(\sigma^2 + \sigma_x'^2)}{(\sigma^2 + \alpha\sigma_x'^2)(\sigma^2 + \sigma_x^2)} + \alpha\left(\frac{\sigma^2 + \sigma_x'^2}{\alpha\sigma_x'^2 + \sigma^2} - \frac{\sigma^2 + \sigma_x^2}{\alpha\sigma_x^2 + \sigma^2}\right)\right) \tag{26}$$

$$+ \frac{\sigma^2(1-\alpha)\|y\|^2}{2}\left(\frac{1}{(\sigma^2 + \sigma_x'^2)(\alpha\sigma_x'^2 + \sigma^2)} - \frac{1}{(\sigma^2 + \sigma_x^2)(\alpha\sigma_x^2 + \sigma^2)}\right). \tag{27}$$

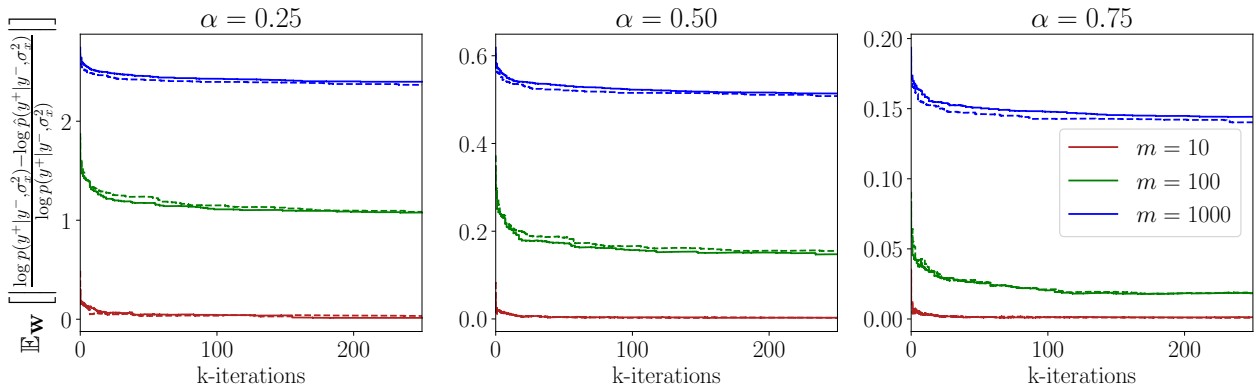

Figure 12. Relative log error between $\widehat{p}(y^+|y^-)$ and $p(y^+|y^-)$ as a function of the number of Monte Carlo integration steps $N$, for different values of $\alpha$ and dimensions $m$ and averaged over 25 samples from $\mathbf{w}$. The full line is obtained by using the analytical posterior law, while the dotted line corresponds to SK-ROCK sampling.

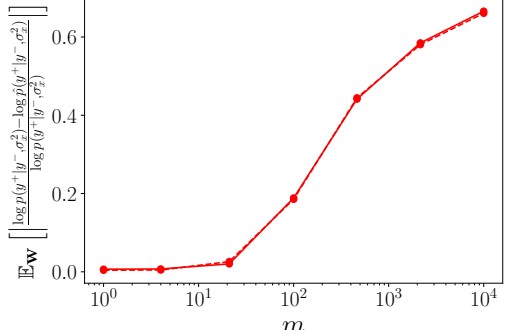

Figure 13. Relative log error between $\widehat{p}(y^+|y^-)$ and $p(y^+|y^-)$ as a function of the dimension, using $N = 50000$ MC steps and averaged over 25 noise realizations, for $\alpha = 0.5$.

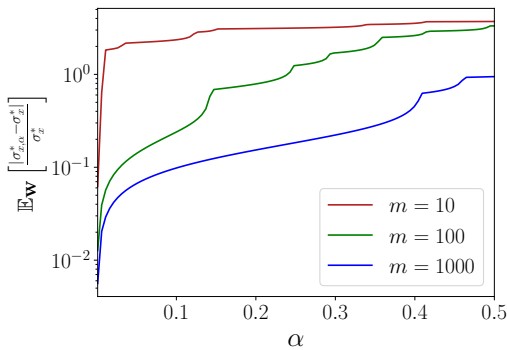

Figure 14. Relative error between $\sigma_x^* = \operatorname{argmax}_{\tilde{\sigma}_x} p(y|\tilde{\sigma}_x)$ and the average of $\sigma_{x,\alpha}^* = \operatorname{argmax}_{\tilde{\sigma}_x} p(y^+|y^-, \tilde{\sigma}_x)$ over 25 noise realizations as a function of $\alpha$.

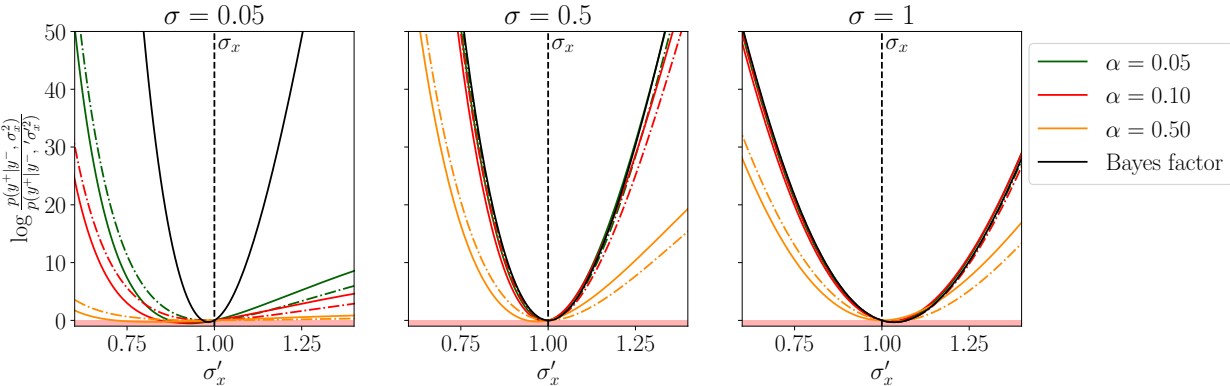

Figure 15. Analytical Bayes Factor $\log \frac{p(y|\sigma_x^2)}{p(y|\sigma_x'^2)}$, and log ratios $\log \frac{p(y^+|y^-, \sigma_x^2)}{p(y|\sigma_x'^2)}$ for different values of $\alpha$ and for different values of the measurement noise $\sigma$. The mean over noise splits $\mathbb{E}_{\mathbf{w}}\left[\log \frac{p(y^+|y^-, \sigma_x^2)}{p(y|\sigma_x'^2)}\right]$ is also depicted with a dotted line.

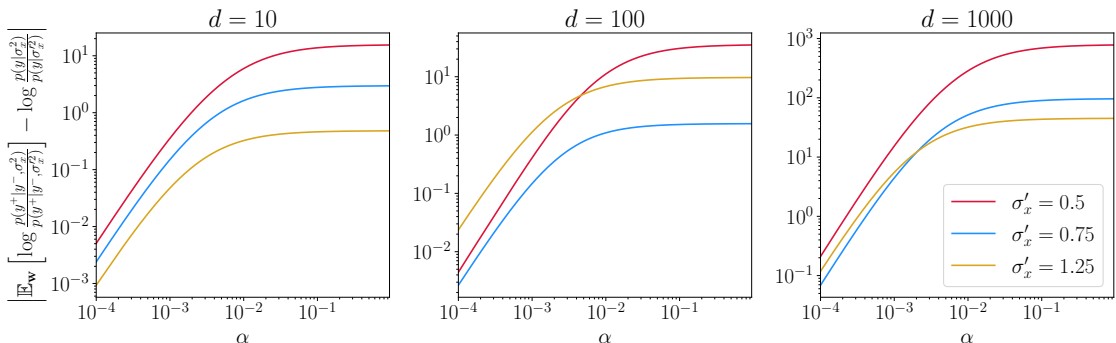

*Figure 16.* Absolute error between the analytical Bayes Factor $\log \frac{p(y|\sigma_x^2)}{p(y|\sigma_x'^2)}$, and mean log ratio $\mathbb{E}_{\mathbf{w}}\left[\log\frac{p(y^+|y^-,\sigma_x^2)}{p(y|\sigma_x'^2)}\right]$ as a function of $\alpha$, for different dimensions $m$, $\sigma = 0.05$. and $\sigma_x = 1$

## C.2. Numerical Experiments

Let $\widehat{p}(y^+|y^-,\sigma_x^2)$ be the approximation of $p(y^+|y^-,\sigma_x^2)$ computed by drawing from the posterior law $x|y^-$, following Eq. (12) of the main paper, either by using the analytical posterior law, or by simulating this distribution with the SK-ROCK algorithm (Abdulle et al., 2018; Pereyra et al., 2020). Fig. 12 represents the relative error between the analytical value and the estimator as a function of the iterations for different values of $\alpha$ and dimensions of the target vector. Full lines correspond to Monte Carlo approximations of the posterior $x|y^-$, while the dotted lines are obtained by drawing from the analytical posterior. Both plots are obtained by averaging the error over 25 samples of $\mathbf{w}$ for $\sigma_x = 1$ and $\sigma = 0.05$. The additional error caused by sampling with SK-ROCK seems negligible relative to the Monte Carlo integration error. Note that the convergence speed is slow, and the approximation error increases the relative error for a fixed number of iterations. Expectedly, the error also increases with the dimension $m$, as depicted in Fig. 13, where we plot the relative error as a function of $m$. We also illustrate the discrepancy between the optimum $\sigma_x^*$ of $p(y|\tilde{\sigma}_x)$ and the optimum $\sigma_{x,\alpha}^*$ of $p(y^+|y^-\tilde{\sigma}_x)$ with respect to $\tilde{\sigma}_x$ in Fig. 14. The error between both values vanishes as $p(y^+|y^-)$ approaches $p(y)$. Finally, we graphically represent in Fig. 15 the behavior of $\log \frac{p(y^+|y^-,\sigma_x^2)}{p(y^+|y^-,\sigma_x'^2)}$ and the analytical Bayes factor $\log \frac{p(y|\sigma_x^2)}{p(y|\sigma_x'^2)}$ for different values of $\alpha$ and measurement noise $\sigma$, for a set ground truth deviation $\sigma_x = 1$. Note that the ratio is more informative when $\sigma$ is low, requiring smaller values of $\alpha$ to become accurate. Fig. 16 illustrates the point-wise convergence of the noise-averaged approximate Bayes factor $\mathbb{E}_{\mathbf{w}}\left[\frac{p(y^+|y^-,\sigma_x'^2)}{p(y^+|y^-,\sigma_x^2)}\right]$ to the analytical Bayes factor as $\alpha$ vanishes to 0, for different values of $\sigma_x'$.

## C.3. Proof of Proposition 1

We give here a proof of the limit $p(y^+|y^-) \to p(y)$ as $\alpha \to 0$.

**Assumption 1.**

1. $\mathbf{w}$ *admits a continuous density* $p_{\mathbf{w}}$ *on* $\mathbb{R}^m$ *that is strictly positive and bounded by* $M_{\mathbf{w}} < \infty$.

2. *For* $p(x)$-*a.e.* $x$, *the likelihood* $y \mapsto p_{\mathbf{y}|\mathbf{x}}(y|x)$ *is continuous on* $\mathbb{R}^m$, *and* $\sup_{x,y} p_{\mathbf{y}|\mathbf{x}}(y|x) \leq M_\ell < \infty$.

**Proposition 2.** *Under Assumption 1, fix* $y \in \mathbb{R}^m$ *such that* $p_{\mathbf{y}}(y) > 0$. *Then, for* $p_{\mathbf{w}}$-*a.e. realization* $w$, *defining* $y^+ = y + c_\alpha w$ *and* $y^- = y - w/c_\alpha$ *with* $c_\alpha = \sqrt{\alpha/(1-\alpha)}$,

$$\lim_{\alpha \to 0^+} p_{\mathcal{M}}(y^+|y^-) = p_{\mathcal{M}}(y).$$

*Proof.* We fix $w$ with $p_{\mathbf{w}}(w) > 0$ and remove the $\mathcal{M}$ subscript for clarity. By Bayes' rule,

$$p(y^+|y^-) = \int p(y^+|x)\, p(x|y^-)dx = \int p(y^+|x)\, \frac{p(y^-|x)}{p(y^-)}\, p(x)\, dx. \tag{28}$$

Since $y^- = y - w/c_\alpha$, conditioning on $\mathbf{y} = u$ gives $y^- = u - w/c_\alpha$, so the change of variables $w \mapsto y^-$ (with Jacobian $c_\alpha$) yields

$$p(y^-|x) = c_\alpha \int p_{\mathbf{y}|\mathbf{x}}(u|x) \, p_{\mathbf{w}}\big(c_\alpha(u - y^-)\big) \, du, \tag{29}$$

and analogously for $p(y^-)$ with $p_{\mathbf{y}|\mathbf{x}}(\cdot|x)$ replaced by $p_{\mathbf{y}}(\cdot)$.

*Step 1:* For any $u$, $c_\alpha(u - y^-) = c_\alpha(u - y) + w \to w$ as $\alpha \to 0^+$. By (A1.1) and continuity, the integrand in Eq. (29) converges pointwise (in $u$) to $p_{\mathbf{y}|\mathbf{x}}(u|x) \, p_{\mathbf{w}}(w)$, with $u$-integrable dominating function $M_{\mathbf{w}} \, p_{\mathbf{y}|\mathbf{x}}(\cdot|x)$. Dominated convergence yields

$$\lim_{\alpha \to 0^+} c_\alpha^{-1} p(y^-|x) = p_{\mathbf{w}}(w), \qquad \lim_{\alpha \to 0^+} c_\alpha^{-1} p(y^-) = p_{\mathbf{w}}(w),$$

and hence, for each $x$, we get

$$\lim_{\alpha \to 0^+} \frac{p(y^-|x)}{p(y^-)} = 1.$$

*Step 2:* From Eq. (29) and (A1.1), $p(y^-|x) \le c_\alpha M_{\mathbf{w}}$ for all $x$. By Step 1 applied to the denominator, there exists $\alpha_0 = \alpha_0(y, w) > 0$ (independent of $x$) such that $p(y^-) \ge c_\alpha \, p_{\mathbf{w}}(w)/2$ for all $\alpha < \alpha_0$. Combining with $p(y^+|x) \le M_\ell$ from (A1.2), for all $\alpha < \alpha_0$ and all $x$,

$$p(y^+|x) \frac{p(y^-|x)}{p(y^-)} p(x) \;\le\; \frac{2 \, M_\ell \, M_{\mathbf{w}}}{p_{\mathbf{w}}(w)} \, p(x),$$

which is $x$-integrable.

*Step 3:* Writing $p(y^+|x) = \int p_{\mathbf{y}|\mathbf{x}}(y^+ - c_\alpha v|x) \, p_{\mathbf{w}}(v) \, dv$, continuity of $p_{\mathbf{y}|\mathbf{x}}(\cdot|x)$, $y^+ \to y$, and dominated convergence (with dominator $M_\ell \, p_{\mathbf{w}}$) give $\lim_{\alpha \to 0^+} p(y^+|x) = p_{\mathbf{y}|\mathbf{x}}(y|x)$ for each $x$. Combining with Step 1, the integrand of Eq. (28) converges pointwise to $p_{\mathbf{y}|\mathbf{x}}(y|x) \, p(x)$. Applying dominated convergence to Eq. (28) with the bound from Step 2, gives

$$\lim_{\alpha \to 0^+} p(y^+|y^-) = \int p_{\mathbf{y}|\mathbf{x}}(y|x) \, p(x) \, dx = p(y). \qquad \square$$

# D. Kernel Selection

## D.1. Implementation Details

We give here some details on the implementation of the kernel selection experiment from the main paper. In all cases, we use the SK-ROCK algorithm to sample the posterior law, using $s = 15$ inner iterations and the potential of the gradient step denoiser (Hurault et al., 2021) as prior. We set the regularization parameter $\lambda$ to 110 for every experiment when computing $\widehat{\Phi}_y^1$, based on a prior study of the reconstruction's quality on a single observation. The standard deviation parameter of the denoiser is also fixed to 0.1.

We re-use the same Markov chain to simulate both the prior and posterior laws in order to apply SAPG (Vidal et al., 2020), adding respectively 15 and 25 thinning iterations before each posterior and prior sample. We initialize the algorithm with the reference value 110, and perform 150 SAPG iterations, generating approximately 6000 samples in the process. While this number could be reduced by increasing the step size, this illustrates a limitation of SAPG, which requires careful per-application tuning to work best, especially when a good first estimate is unavailable. In contrast, we use a single chain to generate the samples in our method, using 20 thinning iterations before swapping to a new noise realization, for a total of approximately 1200 sampling steps. Note also that, as we did not tune the regularization parameter for our method, the prior's misspecification is higher. We use a single V100 16GB GPU to process a single image.

Note that we used a FFT-based blur operator, which involves the application of circular padding. In order to avoid a potential bias due to this padding, we ignore the padding pixels when computing each metric (*i.e.* use "valid" padding). The amount of pixels removed is based on the span of the largest convolutional kernel. We used an implementation based on the Deepinv library (Tachella et al., 2025) for the forward operator. The analytical definition of each kernel is given in Tab. 6 below.

$$\begin{aligned}
\kappa_{\mathcal{G}}(\sigma) \quad &: (x,y) \mapsto e^{-(x^2+y^2)/2\sigma^2} \\
\kappa_{\mathcal{M}}(\sigma,\mu) \quad &: (x,y) \mapsto (\sigma^2(x^2+y^2)/\mu+1)^{-(\mu/2+1)} \\
\kappa_{\mathcal{L}}(\sigma) \quad &: (x,y) \mapsto e^{\sigma(-|x|+|y|)} \\
\kappa_{\mathcal{U}}(s) \quad &: (x,y) \mapsto \mathbb{1}_{x,y \leq s}
\end{aligned}$$

*Table 6.* Unnormalized blurring kernels.

## D.2. Numerical Results

We report in this section the numerical results from the kernel selection experiments. Tab. 7 displays the values of $\widehat{\Phi}_y^1(\mathcal{M})$. Each row corresponds to different measurements generated using different blurring kernels, while the columns correspond to the tested kernels. I1, I2, I3 denote the three test images depicted in the main paper. Tab. 8 gives the average value of the estimator over I1, I2, I3. The estimator fails to select the correct kernel in only two out of fifteen cases when using a single observation and selects the correct kernel in every case when averaging over the three test images. The values of the estimator are quite close with the number of samples used. The single-shot performance might still be improved by increasing the samples in the Monte Carlo approximation. Tab. 9 gives the average (unnormalized) error of the MAP reconstruction, obtained after tuning the regularization parameter by applying SAPG. Even in the few-shot setting, we only reach 60% accuracy.

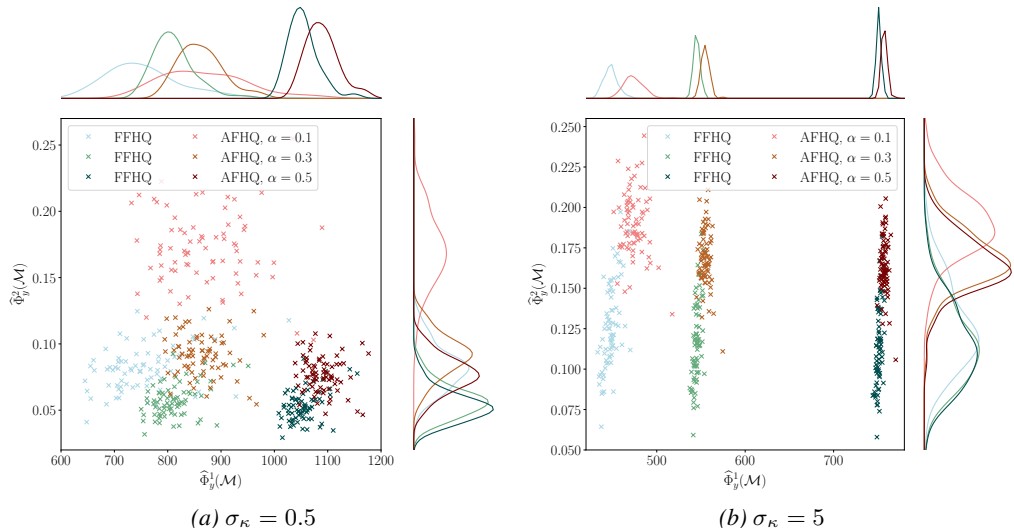

*(a)* $\sigma_\kappa = 0.5$        *(b)* $\sigma_\kappa = 5$

*Figure 17.* Distributions of the values taken by $\widehat{\Phi}_y^1(\mathcal{M})$ and $\widehat{\Phi}_y^2(\mathcal{M})$ over the FFHQ subset, for the FFHQ and AFHQ-trained models, at $\alpha = 0.1, 0.3, 0.5$.

| Test
GT | $\kappa_{\mathcal{G}}(2)$
I1 | I2 | I3 | $\kappa_{\mathcal{M}}(0.5,1)$
I1 | I2 | I3 | $\kappa_{\mathcal{L}}(0.4)$
I1 | I2 | I3 | $\kappa_{\mathcal{U}}(3)$
I1 | I2 | I3 | $\kappa_{\mathcal{G}}(2.5)$
I1 | I2 | I3 |
|---|---|---|---|---|---|---|---|---|---|---|---|---|---|---|---|
| $\kappa_{\mathcal{G}}(2)$ | **45.72** | **58.83** | **56.87** | 46.01 | 60.50 | 57.31 | 51.79 | 67.34 | 59.42 | 49.90 | 65.49 | 58.44 | 48.07 | 61.60 | 59.41 |
| $\kappa_{\mathcal{M}}(0.5,1)$ | 46.50 | 62.43 | 53.96 | **44.32** | **56.76** | **53.64** | 45.55 | 58.81 | 54.48 | 50.10 | 67.69 | 54.80 | 48.14 | 60.16 | 54.95 |
| $\kappa_{\mathcal{L}}(0.4)$ | 41.09 | 55.26 | 45.89 | 39.63 | 54.04 | 44.96 | **38.55** | **52.97** | **44.73** | 42.10 | 56.78 | 45.88 | 39.86 | 54.73 | 45.10 |
| $\kappa_{\mathcal{U}}(3)$ | 46.35 | 60.02 | **53.26** | 48.37 | 62.59 | 53.75 | 51.55 | 68.36 | 55.09 | **45.10** | **57.07** | 53.46 | 47.03 | 58.07 | 54.36 |
| $\kappa_{\mathcal{G}}(2.5)$ | 40.31 | 53.05 | 46.42 | 40.16 | 53.03 | 46.14 | 41.02 | 54.02 | **46.13** | 40.56 | 53.41 | 46.44 | **39.39** | **52.46** | 46.29 |

*Table 7.* Values of $\widehat{\Phi}_y^1 - 1100$ for the three test images for different ground truth blurring kernel (rows), computed using 10 noise realizations with 100 steps each and $\alpha = 0.5$. The best values for each row are highlighted in bold font, with a mean accuracy of 86.7% over the 15 experiments.

| Test
GT | $\kappa_{\mathcal{G}}(2)$ | $\kappa_{\mathcal{M}}(0.5, 1)$ | $\kappa_{\mathcal{L}}(0.4)$ | $\kappa_{\mathcal{U}}(3)$ | $\kappa_{\mathcal{G}}(2.5)$ |
|---|---|---|---|---|---|
| $\kappa_{\mathcal{G}}(2)$ | **13.808** | 14.603 | 19.515 | 17.942 | 16.361 |
| $\kappa_{\mathcal{M}}(0.5, 1)$ | 14.295 | **11.574** | 12.947 | 17.534 | 14.416 |
| $\kappa_{\mathcal{L}}(0.4)$ | 7.416 | 6.210 | **5.414** | 8.252 | 6.566 |
| $\kappa_{\mathcal{U}}(3)$ | 13.213 | 14.904 | 18.335 | **11.875** | 13.155 |
| $\kappa_{\mathcal{G}}(2.5)$ | 6.592 | 6.443 | 7.055 | 6.802 | **6.045** |

*Table 8.* Values of $\widehat{\Phi}_y^1(\mathcal{M}) - 1100$ averaged over three test images for different ground truth blurring kernel (rows), computed using 10 noise realizations with 100 steps each and $\alpha = 0.5$.

| Test
GT | $\kappa_{\mathcal{G}}(2)$ | $\kappa_{\mathcal{M}}(0.5, 1)$ | $\kappa_{\mathcal{L}}(0.4)$ | $\kappa_{\mathcal{U}}(3)$ | $\kappa_{\mathcal{G}}(2.5)$ |
|---|---|---|---|---|---|
| $\kappa_{\mathcal{G}}(2)$ | **14.936** | 14.955 | 19.647 | 22.448 | 22.378 |
| $\kappa_{\mathcal{M}}(0.5, 1)$ | 17.353 | **12.396** | 17.230 | 22.374 | 20.946 |
| $\kappa_{\mathcal{L}}(0.4)$ | 12.418 | 11.050 | **10.991** | 15.277 | 15.731 |
| $\kappa_{\mathcal{U}}(3)$ | **14.515** | 16.809 | 16.773 | 17.737 | 18.355 |
| $\kappa_{\mathcal{G}}(2.5)$ | 10.825 | **9.085** | 10.620 | 12.235 | 12.927 |

*Table 9.* Values of $\|\kappa * \widehat{x}_\kappa - y_{\kappa_{\mathrm{GT}}}\|_2^2$ - 560 averaged over three test images, where $\widehat{x}_\kappa$ denotes the approximate MAP reconstruction using the tested blurring kernel $\kappa$ for the forward model.

# E. Misspecification Detection

## E.1. Additional Observations for the Deblurring Experiment

We provide here some figures to further illustrate the observations made in Section 4.3.1 of the main paper.

Fig. 17 depicts the distributions of $\widehat{\Phi}_y^1(\mathcal{M})$ and $\widehat{\Phi}_y^2(\mathcal{M})$ for the FFHQ and AFHQ-trained models for different values of $\alpha$, at $\sigma_\kappa = 0.5$ and $\sigma_\kappa = 5$. The perceptual variance of the samples greatly increases when we allow $\alpha$ to reduce for the OOD model, while it is less affected for the ID model. Fig. 18 shows that a poor choice of $\alpha$ translates into increased statistical error rates when using $\widehat{\Phi}_y^2(\mathcal{M})$ for model selection. Indeed, when $\alpha$ is close to $0.5$, the noise quantity imbalance between $y^+$ and $y^-$ vanishes, and the perceptual variance between samples is reduced, rendering the task of detecting OOD images from sample variance less effective. Note that when the amount of information available in the measurements is low, the perceptual variance of the samples is high, even for ID images. The variations in specific details of the samples, such as a mouth being open or closed, can cause the test to fail in extreme cases. Fig. 19 shows such an example, where $\widehat{\Phi}_y^2(\mathcal{M})$ is slightly higher for the FFHQ-trained model. Note, however, that $\widehat{\Phi}_y^1(\mathcal{M})$ chooses the correct model in this case. Fig. 20 represents the distributions of $\widehat{\Phi}_y^1(\mathcal{M})$ and $\widehat{\Phi}_y^2(\mathcal{M})$ for the FFHQ and AFHQ-trained models at different blur levels over the FFHQ subset. As the blur level increases, the values of $\widehat{\Phi}_y^2(\mathcal{M})$ spread out, while the distribution of $\widehat{\Phi}_y^1(\mathcal{M})$ becomes sharper. Indeed, at higher blur values, the inter-sample differences are small in the measurement space, while the sample variety increases. Finally, a concern can be raised by observing the low convergence speed of the estimator in the Gaussian analytical case (see Fig. 12). The experiments on natural images show that fine convergence is not required in order to have accurate model selection or misspecification detection. Fig. 21 shows the value of the estimator $\widehat{\Phi}_y^2(\mathcal{M})$ as a function of the number of $x|y^-$ samples. We can observe that, even though the estimator has not converged, the variation with respect to new iterations is negligible. However, in border cases where the information available is low, such as the case depicted in Fig. 19, adding some iterations might improve results.

## E.2. Model Selection: Prior Comparison

In addition to the model selection experiments of Sections 4.2 and E.6, we also perform model selection in the context of the deblurring problem of Section 4.3.1, for $\sigma_\kappa = 0.5$ and $\sigma_\kappa = 2$. We compare the following four models: two diffusion priors trained respectively on FFHQ and AFHQ-dogs (implemented via DiffPIR samplers), a gradient step denoiser prior (implemented via SKROCK sampling) and a total variation norm prior (also implemented via SKROCK sampling). We report the results in Tab. 10 for the FFHQ subset, and Tab. 11 for the 15 ID images of the Celeb subset. The means of the estimators both select the most appropriate model for each dataset, *i.e.*, the FFHQ-trained diffusion model. In practice,

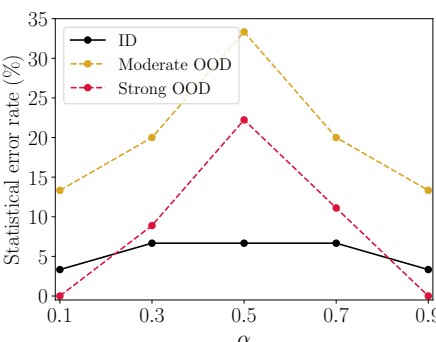

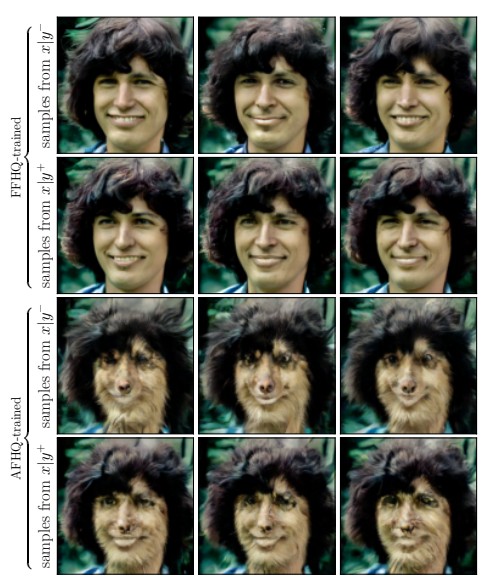

Figure 18. Type 1 error rate on ID images, and type 2 error rates for moderately OOD and strongly OOD images, as a function of $\alpha$ for the deblurring problem with $\sigma_\kappa = 0.5$.

Figure 19. Samples from $x|y^-$ and $x|y^+$ for the FFHQ and AFHQ-trained models, where $y$ is obtained by blurring a FFHQ image with $\sigma_\kappa = 5$.

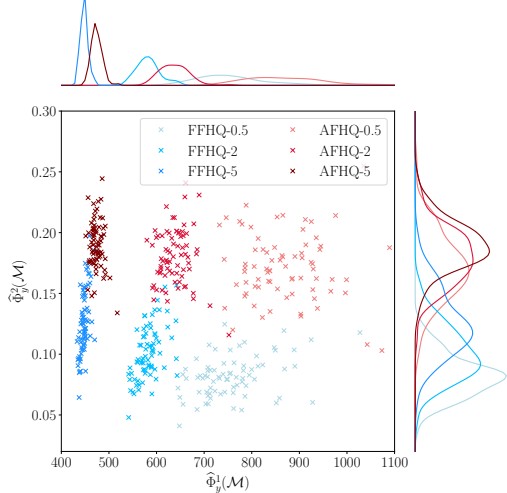

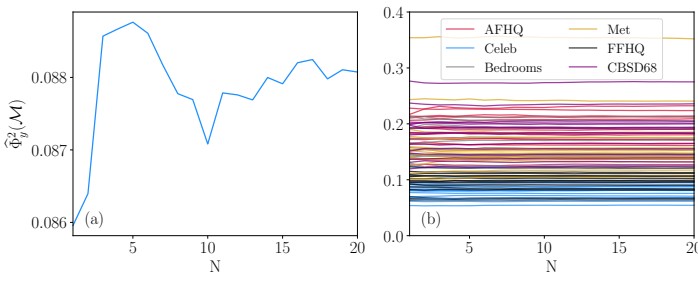

Figure 20. Distributions of the values taken by $\widehat{\Phi}_y^1(\mathcal{M})$ and $\widehat{\Phi}_y^2(\mathcal{M})$ over the FFHQ subset, for the FFHQ and AFHQ-trained models, at $\sigma_\kappa = 0.5, 2, 5$ and $\alpha = 0.1$.

Figure 21. $\widehat{\Phi}_y^2(\mathcal{M})$ as a function of the number of steps $N$, at a fixed number of noise realizations $K = 10$, for a single Celeb-Faces image (a) and for each image of the test dataset (b).

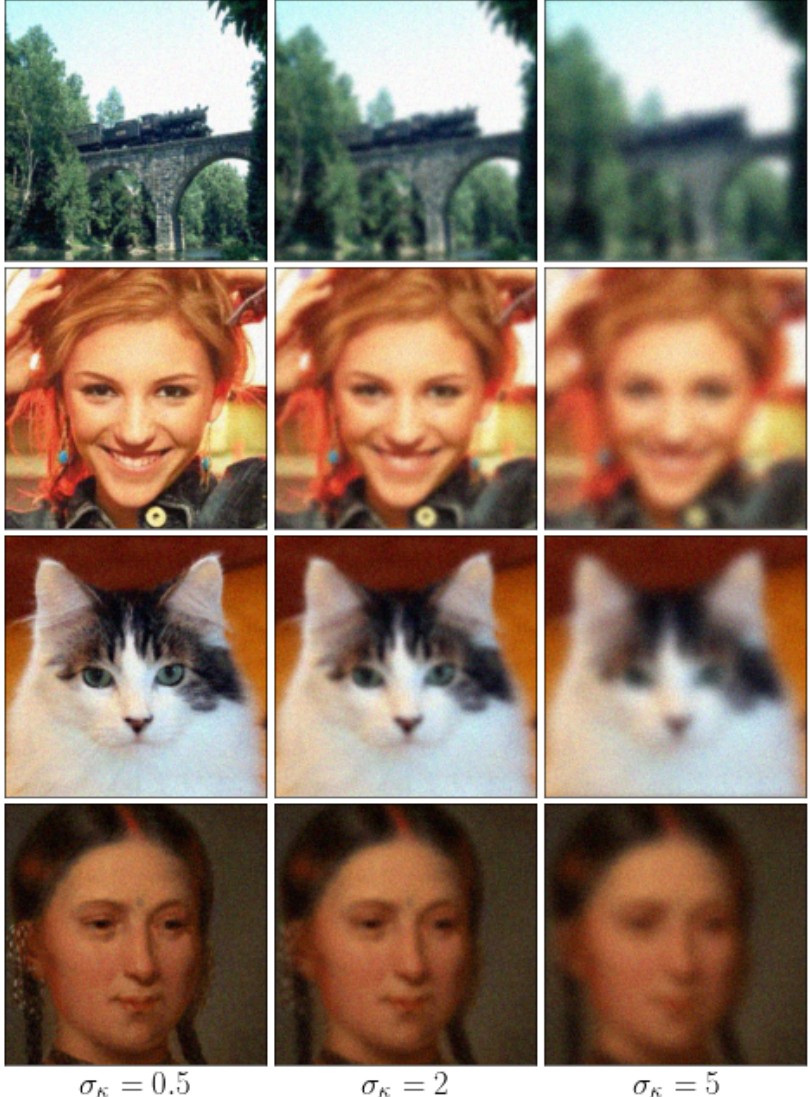

*Figure 22.* Some observation vectors for the deblurring experiment of Section 4.3.1, at the three blurring levels.

the FFHQ-trained diffusion model is also selected for each image separately in a single-shot manner. Note that the four models are also ranked correctly by $\Phi_y^1$ and $\Phi_y^2$, both at high and low blur levels.

| Samplers | $\Phi_y^1(\mathcal{M})\downarrow$ | | $\Phi_y^2(\mathcal{M})\downarrow$ | | PSNR (dB) $\uparrow$ | | LPIPS $\uparrow$ | |
|---|---|---|---|---|---|---|---|---|
| | $\sigma_\kappa = 0.5$ | $\sigma_\kappa = 2$ | $\sigma_\kappa = 0.5$ | $\sigma_\kappa = 2$ | $\sigma_\kappa = 0.5$ | $\sigma_\kappa = 2$ | $\sigma_\kappa = 0.5$ | $\sigma_\kappa = 2$ |
| DiffPIR FFHQ | **757** | **583** | **0.0817** | **0.101** | **36.4** | **32.1** | **0.0534** | **0.0741** |
| DiffPIR AFHQ | 869 | 638 | 0.171 | 0.180 | 35.3 | 30.7 | 0.125 | 0.230 |
| SKROCK GSD | 994 | 721 | 0.223 | 0.332 | 32.4 | 29.3 | 0.247 | 0.247 |
| SKROCK TV | 2320 | 922 | 0.635 | 0.725 | 30.2 | 28.7 | 0.261 | 0.413 |

*Table 10.* Model selection among FFHQ and AFHQ-trained diffusion models used with DiffPIR, and SKROCK samplers with gradient step denoiser and total variation priors on the FFHQ subset for the deblurring problem. The supervised PSNR and LPIPS between the ground truth and the posterior mean computed using 40 posterior samples are reported alongside our unsupervised estimators.

| Samplers | $\Phi_y^1(\mathcal{M})\downarrow$ | | $\Phi_y^2(\mathcal{M})\downarrow$ | | PSNR (dB) $\uparrow$ | | LPIPS $\uparrow$ | |
|---|---|---|---|---|---|---|---|---|
| | $\sigma_\kappa = 0.5$ | $\sigma_\kappa = 2$ | $\sigma_\kappa = 0.5$ | $\sigma_\kappa = 2$ | $\sigma_\kappa = 0.5$ | $\sigma_\kappa = 2$ | $\sigma_\kappa = 0.5$ | $\sigma_\kappa = 2$ |
| DiffPIR FFHQ | **749** | **579** | **0.0819** | **0.0982** | **35.9** | **31.7** | **0.0587** | **0.0845** |
| DiffPIR AFHQ | 841 | 627 | 0.124 | 0.145 | 35.4 | 30.9 | 0.113 | 0.208 |
| SKROCK GSD | 942 | 696 | 0.205 | 0.318 | 32.3 | 29.6 | 0.252 | 0.249 |
| SKROCK TV | 2270 | 901 | 0.633 | 0.709 | 30.2 | 28.8 | 0.255 | 0.418 |

*Table 11.* Model selection among FFHQ and AFHQ-trained diffusion models used with DiffPIR, and SKROCK samplers with gradient step denoiser and total variation priors on the Celeb test subset for the deblurring problem.

### E.3. Choice of Embeddings

In order to assess the sensitivity of the proposed method to a larger choice of embeddings, we carry out an ablation study and compute $\widehat{\Phi}_y^2(\mathcal{M})$ with additional embeddings for the OOD experiments of Section 4.3.1. We test Dinov2 (Oquab et al., 2023) embeddings (non-finetuned), the SOTA perceptual embeddings Dreamsim (Fu et al., 2023), and embeddings extracted from the original patch-32 CLIP (Radford et al., 2021). In the case of CLIP-based embeddings, we extract the output from the second-to-last hidden state of the visual transformer as suggested in (Li et al., 2023) and normalize it by the L2 norm. We report the rejection rate for each subset and embeddings in Tab. 12. We note that the results obtained with Dinov2 are similar to those obtained with LPIPS, highlighting the efficiency of the method when using general-purpose, informative embeddings. CLIP does not produce meaningful results, as the samples produced for this experiment are semantically close. Dreamsim is more discriminative: the reference estimator values on FFHQ are more concentrated for this metric, resulting in a higher rejection rate for ID images. This metric is more sensitive to style changes, which also has the consequence of increasing the power for mildly OOD images, even at higher blur levels. While powerful general-purpose embeddings yield a good accuracy for this task, these results indicate that aggregating the estimators computed from different embeddings or using specialized, task-specific embeddings could further improve OOD detection. These considerations go beyond the scope of this article and will be studied in future work.

| | | $\sigma_\kappa = 0.5$ | | | | $\sigma_\kappa = 2$ | | | | $\sigma_\kappa = 5$ | | | |
|---|---|---|---|---|---|---|---|---|---|---|---|---|---|
| | | LPIPS | DINO | CLIP | DS | LPIPS | DINO | CLIP | DS | LPIPS | DINO | CLIP | DS |
| Type I Error | FFHQ | 0 % | 0 % | 6.7 % | 13% | 6.7 % | 0 % | 0 % | 13% | 0 % | 0 % | 0 % | 13 % |
| | Celeb | 6.7 % | 13% | 0 % | 60 % | 6.7 % | 6.7 % | 6.7 % | 13% | 6.7 % | 6.7 % | 0 % | 40 % |
| Power | Moderate OOD | 87 % | 87 % | 6.7 % | 100% | 73 % | 80 % | 6.7 % | 93 % | 60 % | 67 % | 6.7 % | 87 % |
| | Strong OOD | 100 % | 100 % | 42 % | 100% | 100 % | 100 % | 33 % | 100 % | 100 % | 100 % | 38 % | 100 % |

*Table 12.* Type I error rate (incorrect rejection of ID samples from FFHQ, Celeb) and Power (correct rejection of moderate OOD (Met) and strong OOD (bedrooms, CBSD68, AFHQ) examples), when using LPIPS, Dinov2, CLIP and Dreamsim (DS) to compute $\widehat{\Phi}_y^2(\mathcal{M})$.

|  |  | $\sigma_\kappa = 0.5$ | $\sigma_\kappa = 2$ | $\sigma_\kappa = 5$ |
|---|---|---|---|---|
| Type I Error | FFHQ | 0% | 0% | 0% |
|  | Celeb | 13% | 6.7% | 6.7% |
| Power | Moderate OOD | 87% | 80% | 67% |
|  | Strong OOD | 100% | 100% | 100% |

*Table 13.* Type I error rate (incorrect rejection of ID samples from FFHQ, Celeb) and Power (correct rejection of moderate OOD (Met) and strong OOD (bedrooms, CBSD68, AFHQ) examples), when using the Dinov2 embeddings to compute $\widehat{\Phi}^2_y(\mathcal{M})$.

| **Type I Error** | | **Power** | |
|---|---|---|---|
| FFHQ | Celeb | Moderate OOD | Strong OOD |
| 6.7% | 6.7% | 80% | 100% |

*Table 14.* Type I error rate (incorrect rejection of ID samples) and Power (correct rejection of moderate OOD and strong OOD images for $\sigma_\kappa = 0.5$ and a Poisson noise model.

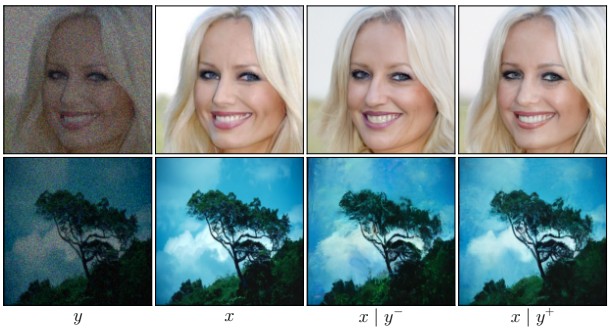

*Figure 23.* Measurements $y$, ground truth image $x$, samples from $x|y^-$ and $x|y^+$ for the Poisson noise experiment, where $y$ is obtained by blurring respectively a CelebA-HQ and CBSD68 image.

### E.4. Poisson Noise

To demonstrate the ability of the method to adapt to different noise models of the exponential family, we consider here the same forward model as in the last section, replacing the additive Gaussian noise with Poisson noise. Poisson noise arises, for example, in photon-starved imaging problems where each component of the measurement $\mathbf{y}$ is modeled as a discrete photon count following a Poisson distribution. The observations are distributed as (Melidonis et al., 2023)

$$\mathbf{y} \mid x_\star \sim \gamma \mathcal{P}\left(\frac{1}{\gamma} A(x_\star)\right), \tag{30}$$

where $\gamma$ is related to the strength of the so-called shot noise of the sensor. We recall the form of the associated negative log likelihood

$$-\log p(\mathbf{y} = y'|x) = \sum_{i=1}^{m} \left([A(x)/\gamma]_i - y'_i \log[A(x)/\gamma]_i + \log(y'_i!)\right), \tag{31}$$

with $y' = y/\gamma$. The data fission strategy for the Poisson distribution is derived from (Monroy et al., 2025) and is given by

$$\begin{aligned} \mathbf{y}^+ = f_\alpha^+(\mathbf{y}, \mathbf{w}) &:= \frac{\mathbf{y} - \gamma \mathbf{w}}{1 - \alpha}, \\ \mathbf{y}^- = f_\alpha^-(\mathbf{y}, \mathbf{w}) &:= \frac{\gamma \mathbf{w}}{\alpha}, \end{aligned} \tag{32}$$

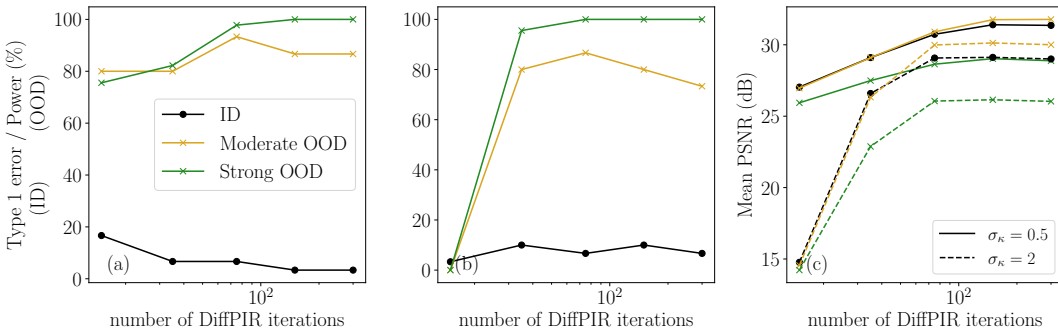

*Figure 24.* Type 1 error (incorrect rejection of ID) and Power (correct rejection of moderate OOD and strong OOD) over the test dataset as a function of the number of discretization steps in DiffPIR, for (a) $\sigma_\kappa = 0.5$ and (b) $\sigma_\kappa = 2$. (c) reports the PSNR obtained by comparing the ground truth with the empirical $\mathbb{E}_\mathbf{w}\left[\mathbb{E}(x|y^-)\right]$.

where $\mathbf{w} \sim \mathrm{Bin}(\mathbf{y}/\gamma, \alpha)$ follows a binomial distribution. The conditional distributions of $\mathbf{y}^+$ and $\mathbf{y}^-$ are given by

$$
\begin{aligned}
\mathbf{y}^+|x_\star &\sim \frac{\gamma}{1-\alpha}\mathcal{P}\left(\frac{(1-\alpha)}{\gamma}x_\star\right), \\
\mathbf{y}^-|x_\star &\sim \frac{\gamma}{\alpha}\mathcal{P}\left(\frac{\alpha}{\gamma}x_\star\right).
\end{aligned}
\tag{33}
$$

When considering Gaussian noise, the proximal operator can be computed explicitly through the SVD decomposition of the blur operator. This is not the case for Poisson noise, for which no explicit formula is known. Following the Prox-DiffPIR method proposed in (Melidonis et al., 2025), we use constrained optimization to compute the proximal operator for the guidance term in DiffPIR. Rather than using LBFGS-B, we apply the PIDAL algorithm (Figueiredo & Bioucas-Dias, 2010), which is based on the ADMM algorithm to enforce positivity constraints.

We perform the same experiment as in Section 4.3.1 with $\sigma\kappa = 0.5$ and $\gamma = 0.05$. We reuse the same pre-trained networks as in the aforementioned Section. We set the maximum number of PIDAL iterations to 1000, stopping optimization early if the relative norm $L_2$ between successive values is less than $10^{-4}$. The parameter $\mu$ is set to 20, and we keep 300 Prox-DiffPIR iterations per sample. Fig.23 represents observations and samples from $x|y^-$ and $x|y^+$ for an ID and an OOD image. Tab. 14 reports the OOD detection results for this experiment. We obtain results similar to those obtained for Gaussian additive noise at the same blur level (see Tab.2), with noisier measurements.

### E.5. Robustness to Sample Degradation

In this section, we highlight the robustness of our estimators to the degradation of the samples' quality. To this end, we evaluate the accuracy of OOD detection when decreasing the number of DiffPIR diffusion steps (iterations). Fig. 24 displays the type 1 error and power of the OOD test based on $\Phi_y^2(\mathcal{M})$ over the test dataset. The ID subset is composed of the 15 test images from FFHQ and the CelebA images. As sample quality decreases, strongly OOD images tend to be rejected less often, especially for $\sigma_\kappa = 0.5$. This is due to the fact that reconstruction quality remains broadly good for OOD images, making the task of detecting OOD samples harder in the presence of additional reconstruction noise. Despite the presence of artifacts and overall low sample quality, the estimator still behaves coherently at 35 iterations for both blur levels. Note that the sampler completely breaks down for $\sigma_\kappa = 2$ and 15 iterations, as can be seen in the PSNR plot of Fig. 24(c). In this case, our estimators also fail, as they rely on posterior samples. Such catastrophic failure cases would require a different approach.

### E.6. MRI Reconstruction

We give here additional illustrations and details for Section 4.3.2 of the main paper. The forward model for the single-coil accelerated MRI problem writes:

$$
y = M\mathcal{F}x,
\tag{34}
$$

|  | $R = 4$ | | $R = 8$ | |
|---|---|---|---|---|
|  | $\widehat{\Phi}_y^2$ | $\widehat{\Phi}_y^1$ | $\widehat{\Phi}_y^2$ | $\widehat{\Phi}_y^1$ |
| Brain | 86% | 60% | 92% | 76% |
| Knee | 88% | 96% | 82% | 96% |

*Table 15.* Accuracy of model selection on the brain and knee scan datasets using $\widehat{\Phi}_y^2$ and $\widehat{\Phi}_y^1$.

where $\mathcal{F}$ denotes the 2D Fourier transform, and $M$ is the sub-sampling operator that applies a mask to the Fourier observations. For simplicity, we do not consider coil sensitivity matrices. In practical experiments, the observations have a fixed under-sampling at low frequencies, and random Gaussian under-sampling at high frequencies. As in the previous section, we use an implementation based on Deepinv (Tachella et al., 2025) to apply the DiffPIR algorithm.

We perform single-shot model selection by computing the estimators on both datasets using both models. The accuracy of each estimator's prediction is reported in Tab. 15. $\widehat{\Phi}_y^2(\mathcal{M})$ performs better than $\widehat{\Phi}_y^1(\mathcal{M})$ when comparing the models on brain images, but fares worse on knee images. This can be partially explained by the fact that the knee model seems slightly under-trained and sometimes produces low-quality knee samples. Fig. 25a displays an image for which $\widehat{\Phi}_y^2(\mathcal{M})$ incorrectly favors the brain model over the knee model, while $\widehat{\Phi}_y^1(\mathcal{M})$ selects the correct model. The brain-trained model hallucinates brain features in its samples, but the perceptual quality of these reconstructions still ranks higher than the knee-trained model's samples. Fig. 25b gives an example of a brain scan for which both estimators select the correct model. Some of the brain's features are recovered by the knee-trained model in samples from $x|y^+$, but are lost in samples from $x|y^-$ due to the added noise. The overall lower performance of $\widehat{\Phi}_y^2(\mathcal{M})$ can also be explained by the fact that the perceptual metric was trained on natural images, and fine-tuning this metric on MRI images might improve results.

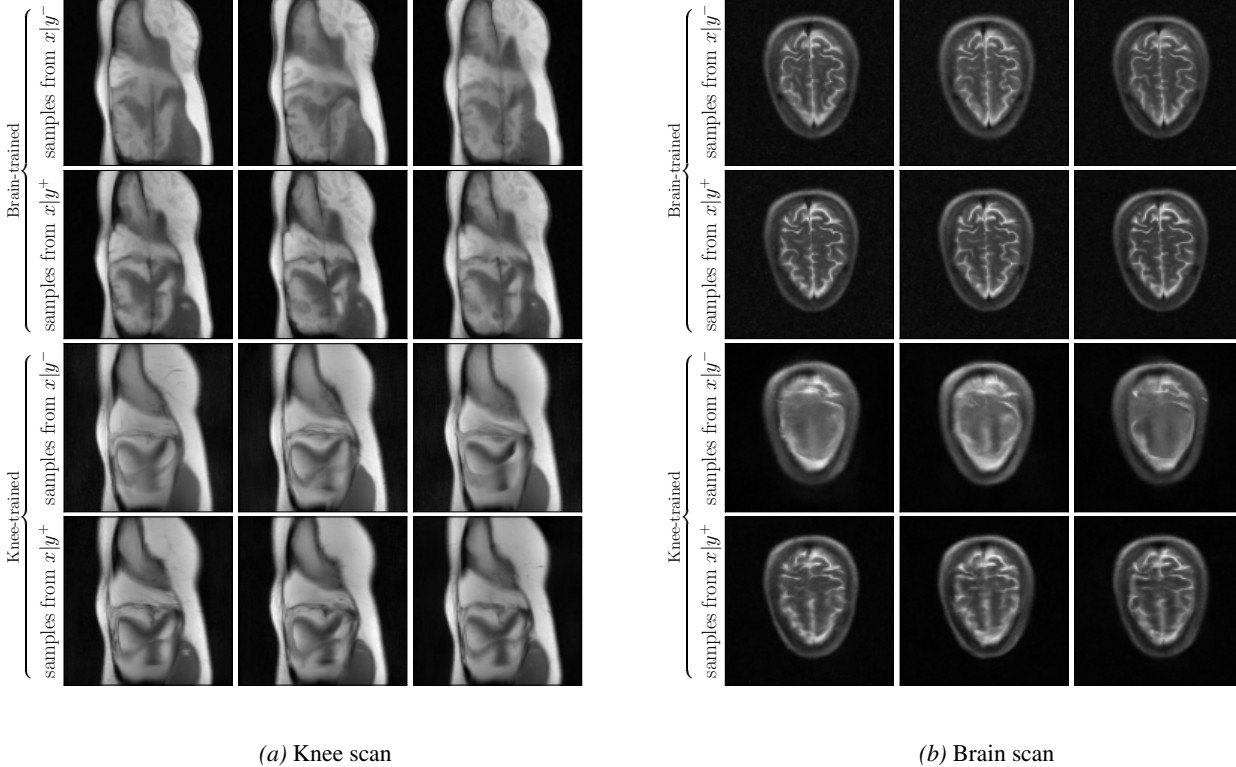

*(a)* Knee scan
*(b)* Brain scan

*Figure 25.* Samples from $x|y^-$ and $x|y^+$ for the brain and knee-trained models, where $y$ is an under-sampled scan with $R = 4$.

