# OpenReview forum: "Bayesian model selection and misspecification testing in imaging inverse problems only from noisy and partial measurements"
_ICML.cc/2026/Conference — ICML 2026 regular_

### Official Review · Reviewer_N6pG · 2026-03-04

**Soundness:** 3
**Presentation:** 3
**Significance:** 3
**Originality:** 4
**Overall Recommendation:** 5
**Confidence:** 4

**Summary:**

In this article the authors propose a novel method for Bayesian model selection in inverse imaging problems. The method only requires a single observation $y \sim P(A(x_*))$. Effectively the method creates additional, synthetic samples $y^+ = y+c_\alpha w$ and $y^-=y-w/c_\alpha$ with $c_\alpha = \sqrt{\alpha/(1-\alpha)}$ and $w$ zero mean Gaussian. These RV are independent conditionally on the underlying $x_*$. The synthetic samples are used to estimate/evaluate a scoring function which evaluates in some sense how well a reconstruction of $x$ from $y^-$ fits also to the observation $y^+$ under the model $\mathcal{M}$. The authors moreover explain that as $\alpha\rightarrow 0$ the proposed scoring approach leads to posterior predictive control which is arguably the ideal method for model selection but suffers from computational burden. The use-cases of the proposed method are two-fold as far as I have understood: Either one may compare a discrete number of different models via their performance on the scoring function to choose the most likely correct one (section 4.2), or one may calibrate the score function value on a sample of data fitting the model which then allows to say in absolute terms of a new given observation is in line with the model (section 4.3).
Finally, the authors provide an extensive set of numerical experiments showcasing the results of the proposed method.

**Compliance With Llm Reviewing Policy:**

Affirmed.

**Final Justification:**

The tackled problem of model selection/misspecification is interesting and relevant and there exist very limited solutions to this problem.  The paper provides a rather clever, yet easy to understand solution which in my opinion is a strong contribution.

**Key Questions For Authors:**

1. I would like if the authors could turn (11) into a more formal result. I suspect under reasonable assumptions on the densities this should be possible using some type of dominated convergence or a similar technique? (For the Gaussian setting this is done in section A.1 anyway)

2. Eq (12) is unclear to me. First of all I would argue that (12) is the Monte-Carlo estimate of (10) rather then (11). (Of course (10) approximates (11) but only for small $\alpha$ whereas (12) approximates (10) for any alpha). However, on the left in (12) we have a function in $y^+$, $y^-$. On the right these do both not appear anymore. Instead on the left different realizations of $y^+$, $y^-$ are used. If the goal is to approximate (10), in (12) there should probably only be a sum over n indicating the samples of $x|y^-$ for which $y^-$ should be fixed. But I suspect the authors wanted to approximate also already the expectation over $w$, that is (4). In this case, the left-hand side of (12) should be changed.

3. As a smaller suggestion for an adaptation: I think the proposed scoring system would fit very well in conjunction with conformal prediction if I am not mistaken. This would only be a small tweak in the final application of the scoring and lead to theoretically guaranteed OOD "ratings" I believe.

4. Can the authors elaborate a little bit on line 704. I do not completely understand what the sentence is supposed to say and in general the relation to SAPG. As I have understood the referenced SAPG methods refers to a technique for computing the maximum likelihood estimator of a continuous parameter (which may be a parametrization of a Bayesian model). How does this relate to the proposed method? To my understanding in the proposed experiments a discrete set of kernels is compared? It seems in table 7 however the SAPG is used to fit the regularization parameter which is otherwise fixed according to line 700? In this context I also did not really understand the purpose of table 7?

Smaller remarks:
* Notation of $\Phi$ is inconsistent. Sometimes a subscript $y$ is added (eqs. 7,8,9) sometimes it is not added (eqs. 4,5,6)
* In figure 7 I think the vertical lines for the percentiles are missing

**Limitations:**

yes

**Strengths And Weaknesses:**

Soundness: The theoretical elaborations within the paper are technically sound. In my opinion the numerical experiments could be improved a bit. I appreciate that the numerical experiments managed to thoroughly investigate the proposed method from many different angles. However, I was wondering, why the comparison methods (Vidal et al., 2020; Vidal et al., 2021) were only taken into account in section 4.2 but not the other experiments? I would suggest adding comparison methods more often.

Presentation: I found the presentation of the paper good. There is maybe one point I would suggest: It would have been helpful in my opinion to clearly emphasize already before the numerical experiments section (probably at the end of section 3.1) how the final method works maybe in the form of a pseudocode.

Significance:
The paper addresses a relevant problem in inverse imaging problems. I would argue that especially the application of the method to OOD is relevant. Modern Bayesian imaging revolves mostly around designing/using learned priors. However, this intrinsically relies on a match between the prior distribution that has been learned and the distribution the observation is from. Therefore, I would argue that effective methods of detecting a mismatch in this aspect are significant.

Originality:
In my opinion the proposed method is simple/easy to understand, yet creative!

---

> ### Author Rebuttal · Authors · 2026-03-31
>
> We thank the reviewer for their insightful comments, constructive feedback and interest in our method.
>
> **Strengths And Weaknesses**
> 1.
> *I was wondering, why the comparison methods (Vidal et al., 2020; Vidal et al., 2021) were only taken into account in section 4.2 ?*
>
> We thank the reviewer for this remark and agree that this aspect of our comparisons could have been clarified. The problem we consider, high-dimensional model selection and misspecification testing, lies largely beyond the scope of existing methodology, leaving very few meaningful competitors for comparison.
>
> Our approach operates on a single measurement, without ground truth, is broadly applicable to all Bayesian imaging models from which posterior samples can be drawn, scales to high dimensions, and supports both model selection and misspecification detection.
>
> A natural comparator for model selection would be the Bayesian factor between the considered models, but this is computationally intractable in the considered high-dimensional settings without significantly restricting the class of models considered. In particular, the method of Vidal et al. is only suitable for models that are log-concave (so not applicable to diffusion models, for example), hence why it was only considered in Section 4.2.
>
> Unfortunately, there are very limited methods available for comparison, particularly in the considered case of a single measurement and no ground truth data. Following the reviewer’s feedback, we will clarify this limitation in the final manuscript, both in the literature review (Section 1) and in the presentation of the experiments (Section 4).
>
> Following the remarks of Reviewer Dhic, we will add a model selection experiment on a set image restoration task. We will strive to include a comparison of our method with the likelihood-based method of Section 4.2 (without the SAPG component) for this experiment.
>
> 2.
> *It would have been helpful in my opinion to clearly emphasize [...] how the final method works maybe in the form of a pseudocode.*
>
> We thank the reviewer for their suggestion. We will include a pseudo-code to clarify the method before the experiment section.
>
> **Key questions**
>
> 1.
> We thank the reviewer for their suggestion. We will formalize Eq (11) and present the proof in the final supplementary material.
>
> 2.
> We thank the reviewer for spotting these errors, we were incorrectly referencing (11) instead of (10) and incorrectly averaging over $k$, our apologies. As stated at the beginning of the subsection, only one realization of $w$ is used here, and Eq. (12) should read: $\hat{p}\_{\mathcal{M}}(y^+ \vert y^-) = \frac{1}{N}\sum_{n=1}^N p_{\mathcal{M}}(y + c_\alpha w|  x_{n})$
> where $x_{n}$ follows the posterior $ \mathbf{x} \mid y - w/c_\alpha, \mathcal M$ and $w$ is a realization of $\mathcal{N}(0, \sigma^2 I_{m})$. We will fix these issues in the revised manuscript.
>
>
> 3.
> This is an excellent suggestion and we are thankful to the reviewer for noticing this. We will incorporate a conformalized version of our method to the final manuscript.
>
> 4.
> SAPG is used in Vidal et al. (2021) to automatically calibrate the regularization hyper-parameter of the prior underpinning each model under comparison. This calibration is performed by maximum marginal likelihood estimation, hence maximizing the Bayesian evidence of each model, for a fixed choice of blur kernel. This will be clarified in the revised manuscript.
>
> **Additional comments**
>
> We thank the reviewer for noticing the discrepancies in the notations for $\Phi$, and we have corrected them in the revised manuscript. The figures have also been modified.

---

> > ### Author Rebuttal · Reviewer_N6pG · 2026-04-01
> >
> > Authors resolved all my questions

---

### Official Review · Reviewer_Dhic · 2026-03-10

**Soundness:** 2
**Presentation:** 2
**Significance:** 2
**Originality:** 2
**Overall Recommendation:** 4
**Confidence:** 3

**Summary:**

The paper proposes a general methodology for unsupervised model selection and misspecification detection in Bayesian imaging sciences. The approach is based on a novel combination of Bayesian cross-validation and data fission from noisy measurements. The method is compatible with any Bayesian imaging sampler, including diffusion and plug-and-play samplers. Two scoring rules are utilized to perform the Bayesian model selection and misspecification testing.

**Compliance With Llm Reviewing Policy:**

Affirmed.

**Final Justification:**

Since the authors did not respond to my follow-up questions, I am unable to adequately assess the practical significance and real-world applicability of the proposed method. Therefore, I maintain my original score.

---
update

Thank you for the timely response. I have accordingly raised my score.

**Key Questions For Authors:**

Please refer to the weakness listed above.

**Strengths And Weaknesses:**

Strengths:
1. The model selection and misspecification testing are conducted using only noisy measurements in an unsupervised manner, which has promising applications in real-world imaging tasks.
2. A toy Gaussian model and an imaging deblurring problem are considered to evaluate the proposed method.
3. The application to detect the out-of-distribution data is also investigated, which is of great interest for deep learning-based priors.

Weakness:
1. There is no experiment demonstrating model selection among different methods for a fixed image restoration task.
2. The computation cost is not provided, as the cost is a major factor when performing model selection in real-world applications.

---

> ### Author Rebuttal · Authors · 2026-03-31
>
> We thank the reviewer for their comments and interest in our method.
>
> **Weaknesses**
>
> 1.
> *There is no experiment demonstrating model selection among different methods for a fixed image restoration task.*
>
> We appreciate the reviewer’s suggestion. To clarify, Section 4.2 focuses on model selection across different models for a fixed image restoration task. In this setting, all models share the same prior (the gradient-step denoiser of Hurault et al.), while differing in their likelihood functions. The advantage of this setting, as opposed to varying priors, is that the ground-truth likelihood is exactly known, which simplifies the assessment of model selection accuracy.
>
> Based on the reviewer's feedback, we are carrying out an additional image deblurring experiment where we compare different image priors with the same likelihood function, namely a plug-and-play prior, a score-based diffusion prior, and a classical wavelet regularization prior. We will include this additional experiment in the final supplementary material. Because none of these priors is perfectly accurate, in this experiment we assess model selection accuracy by comparing the results of our proposed method with the method selected in a supervised manner by computing PSNR and LPIPS w.r.t. to the true image. This will showcase a realistic scenario where a practitioner uses our method to inform the choice of the prior. Preliminary results suggest that our proposed approach has a model selection accuracy of the order of  $100$ % ($85$ %) in the considered image deblurring setting with the lowest blur level, relative to choosing the model achieving the best PSNR (LPIPS) score in a supervised manner.
>
> 2. *The computation cost is not provided, as the cost is a major factor when performing model selection in real-world applications.*
>
> Please refer to our response to comment 1. of reviewer BCTd, where we provide a discussion on the computational cost of our method in comparison to alternatives and we provide concrete examples of the computational cost. We will add a section in the appendix to discuss the computing time of our method.
>
> **Additional comments**
> We thank the reviewer for their constructive feedback and hope that the additional experiments and clarifications provided in this rebuttal address your concerns. If so, we would be grateful if you could consider revisiting your score in light of the revisions.

---

> > ### Author Rebuttal · Reviewer_Dhic · 2026-04-03
> >
> > Thank you for your response. The reviewer would like clarification on how to determine which prior is most appropriate for a given problem, particularly when multiple priors are available.

---

> > > ### Author Response · Authors · 2026-04-08
> > >
> > > When multiple priors $g_1, \ldots, g_s$ are available, we can select the most appropriate one by computing the estimators $\Phi^1_y(\mathcal M_{g_s})$ and $\Phi^2_y(\mathcal M_{g_s})$ for the corresponding models $ M_{g_s}$ and choosing the model/prior that yields the lowest value.
> > >
> > > For illustration, we computed the values of $\Phi^1_y(\mathcal M_{g_s})$ and $\Phi^2_y(\mathcal M_{g_s})$ over $75$ images extracted from FFHQ, for the following four models: two diffusion priors trained respectively on FFHQ and AFHQ-dogs (implemented via DiffPIR samplers), a gradient step denoiser prior (implemented via SKROCK sampling) and a total variation norm prior (also implemented via SKROCK sampling). We report below the mean values of $\Phi^1_y(\mathcal M)$ and $\Phi^2_y(\mathcal M)$ for these four models for the image deblurring problem of Section 4.2. We also report the (supervised) PSNR and LPIPS metrics that we computed using the approximate posterior mean $\mathbb E \left[x|y\right]$ and the ground truth $x$, which would be unavailable in a real use case.
> > >
> > > For $\sigma_\kappa=0.5$, we get:
> > > Method|$\Phi^1_y(\mathcal M)$ $\downarrow$| $\Phi^2_y(\mathcal M)$ $\downarrow$|PSNR $\uparrow$| LPIPS $\downarrow$
> > > |-|-|-|-|-|
> > > DiffPIR (FFHQ)|**757**|**0.0817**|**36.4**|**0.0534**
> > > DiffPIR (AFHQ)|869|0.171|35.3|0.125
> > > GSD|994|0.223|32.4|0.247
> > > TV|2320|0.635|30.2|0.261
> > >
> > > and for $\sigma_\kappa=2$:
> > >
> > > Method|$\Phi^1_y(\mathcal M)$ $\downarrow$| $\Phi^2_y(\mathcal M)$ $\downarrow$|PSNR $\uparrow$| LPIPS $\downarrow$
> > > |-|-|-|-|-|
> > > DiffPIR (FFHQ)|**583**|**0.101**|**32.1**|**0.0741**
> > > DiffPIR (AFHQ)|638|0.180|30.7|0.230
> > > GSD|721|0.332|29.3|0.247
> > > TV|922|0.725|28.7|0.413
> > >
> > > Observe that $\Phi^1$ and $\Phi^2$ rank the priors correctly. In fact, in this case, both $\Phi^1$ and $\Phi^2$ correctly select the best prior (the FFHQ-trained diffusion model) in each of the $75$ images, achieving excellent accuracy for single-shot unsupervised model detection. Moreover, we obtain this same excellent accuracy for the $15$ images from the Celeb A dataset, which we consider to be in-distribution.
> > > As expected, the diffusion models produce higher quality samples than the other priors, which is clear from the supervised metrics. The differences between the diffusion models are less obvious, especially for the easiest problem $\sigma_\kappa=0.5$ for which the impact of the prior is less noticeable as the likelihood is more informative.
> > >
> > > We hope that the clarifications and additional results provided in this rebuttal address your concerns. If so, we would be grateful if you could consider revisiting your score in light of the revisions.

---

### Official Review · Reviewer_pgjk · 2026-03-10

**Soundness:** 2
**Presentation:** 1
**Significance:** 2
**Originality:** 2
**Overall Recommendation:** 3
**Confidence:** 1

**Summary:**

The present work combines data fission with Bayesian cross-validation for model evaluation in inverse problems.

**Compliance With Llm Reviewing Policy:**

Affirmed.

**Final Justification:**

Some of my concerns persist. Given my expertise, I am not in a position to confidently support either the manuscript in its current form or my initial review.

**Key Questions For Authors:**

Could the authors provide additional comparisons with other model selection or model evaluation methods?

**Limitations:**

yes

**Strengths And Weaknesses:**

**Soundness.** This work introduces a novel, technically sound methodology, which is assessed empirically but not supported by a theoretical analysis. Quite diverse numerical experiments are presented. My main concern is that the experimental evaluation includes only a few competing methods. Stronger comparisons would strengthen the claims made by the authors. So far, the results tend to convince that the proposed method works, but it remains difficult to assess its relative advantage.

**Presentation.** The manuscript contains many stylistic choices sounding overly dramatic. In my opinion, the background section remains somewhat generic and does not sufficiently explain to unfamiliar readers how the proposed method fills a gap in the existing literature. Several notations are not properly introduced, and, when introduced, are sometimes used in ways that do not follow their initial definition. As a result, the paper is very difficult to read, which has unfortunately made it harder to properly assess the contribution of the work.

**Significance.** It tackles a particularly relevant problem since, in many settings, the ground truth is unavailable.

**Originality.** While the individual building blocks of the method are known, their combination resulting in the proposed approach appears to be original.

---

> ### Author Rebuttal · Authors · 2026-03-31
>
> We thank the reviewer for their feedback.
>
> **Strengths And Weaknesses**
>
> 1.
> *This work introduces a novel [...] methodology, which is [...] not supported by a theoretical analysis.*
>
> While we agree that some aspects of the theoretical underpinning could be elaborated further, we emphasize that all core components of the proposed methodology are individually well established. In particular, there is extensive theory on data fission for measurement splitting (e.g., Leiner et al., 2023), proper scoring rules (e.g., Gneiting and Raftery, 2007), Bayesian cross-validation (e.g., Gelman et al., 2013), and the accuracy of Monte Carlo estimators (e.g., Robert and Casella, 2004).
>
> Our contribution lies in integrating these established elements into a unified framework for model assessment in imaging inverse problems. Also, note that the Bayesian model evidence arises as a limiting case of our more general approach, clarifying our connection to classical Bayesian decision theory.
>
> 2.
> *The experimental evaluation includes only a few competing methods [...]. It remains difficult to assess its relative advantage.*
>
> We thank the reviewer for this remark and agree that this aspect of our comparisons could have been clarified. The problem we consider, high-dimensional model selection and misspecification testing, is underserved by existing methodology, leaving very few meaningful competitors for comparison.
>
> Our approach operates on a single measurement, without ground truth, is broadly applicable to all Bayesian imaging models from which posterior samples can be drawn, scales to high dimensions, and supports both model selection and misspecification detection.
>
> A natural comparator for model selection would be the Bayesian factor between the considered models, but this is computationally intractable in the considered high-dimensional settings without significantly restricting the class of models considered (e.g., log-concave models that can be addressed by proximal nested sampling or with the approach of Vidal et al. (2021), which we used for comparison included in Section 4.2).
>
> Similarly, existing out-of-distribution (OOD) detection methods suitable for imaging sciences are often tailored to specific model classes, require ground truth data, or do not scale well to high dimensions. Many also focus on comparing models, rather than assessing whether a single model operates in an OOD regime. To the best of our knowledge, no alternative method enables reliable OOD detection for a single model without using ground truth.
>
> 3.
> *The manuscript contains many stylistic choices sounding overly dramatic.*
>
> We will revise the manuscript to adopt a more measured tone. Could the reviewer provide specific examples to ensure we address these cases effectively?
>
> 4.
> *The background section [...] does not sufficiently explain [...] how the proposed method fills a gap in the existing literature.*
>
> We will revise the introduction to ensure that the background section is clear. We will focus on the "Unsupervised Bayesian model selection" paragraph to explain why Bayesian evidence is unsuitable for modern imaging models. We will also clarify why standard Bayesian cross-validation is not applicable in our setting. Finally, we will emphasize that up to the best of our knowledge, there is no alternative method that is able to do OOD detection in this setting.
>
> 5.
> *Several notations are not properly introduced, and, when introduced, are sometimes used in ways that do not follow their initial definition.*
>
> We will go through the article to ensure the notation is properly introduced and consistent. Please see our response to reviewer N6pG, where we addressed some minor notation issues. We would be grateful if the reviewer could point to specific instances where the notation is unclear or inconsistent?
>
> *Key Questions For Authors:*
>
> 6.
> *Could the authors provide additional comparisons with other model selection or model evaluation methods?*
>
> We refer the reviewer to our response to their second comment. Model selection and misspecification diagnosis is still largely unexplored in imaging science, and there are very limited methods available for comparison. Is the reviewer aware of other methods that we could have used for comparison in our setting ?
>
> **Additional comments**
>
> We thank the reviewer for their constructive feedback and hope that the clarifications provided in this rebuttal address your concerns. If so, we would be grateful if you could consider revisiting your score in light of the revisions.

---

> > ### Author Rebuttal · Reviewer_pgjk · 2026-04-02
> >
> > Thank you for the detailed response. As I am not an expert in this specific area, my assessment should be viewed as an educated guess. Nevertheless, even if the proposed method is claimed to have no direct competitors in high-dimensional settings, it would still be valuable to include some kind of experiments in lower-dimensional regimes where comparisons with established baselines are possible.

---

> > > ### Author Response · Authors · 2026-04-08
> > >
> > > We thank the reviewer for the response. We agree that a comparison with an established baseline is valuable. We have therefore included the following two additional experiments.
> > >
> > > The first comparison is with a state-of-the-art proximal-based nested sampling algorithm, Proxnest, that estimates the Bayesian model evidence. The method was developed in Cai et al. 2022 and extended to deep learning priors in McEwen et al. 2023. We adopt the denoising experimental setting of McEwen, which compares a range of image priors for denoising galaxy images. That article compares a conventional wavelet sparsity prior with a pretrained DnCNN denoiser. We extend the comparison by adding a more complex DRUNet denoiser. For each one of these three models, we compute the log evidence (logZ) using the Proxnest algorithm, the two statistics $\Phi_1$ and $\Phi_2$ using our method, and two supervised model comparison metrics (the PSNR of the posterior mean of $x \mid y^-$ by reusing the generated samples for the $\Phi$ statistics, and the PSNR of the maximum-a-posteriori (MAP) computed with an appropriate optimization algorithm). These supervised metrics are included herein for validatory purposes, as in a real scenario one would not have access to the ground truth image $x$.
> > >
> > > | Prior ($\mathcal M$)   | $\Phi^1_y(\mathcal M)$ ($\downarrow$)  | $\Phi^2_y(\mathcal M)$ ($\downarrow$)  | PSNR $\mathbb{E}(x\|y^-)$ ($\uparrow$) [dB] | MAP PSNR ($\uparrow$) [dB] | logZ ($\uparrow$)  |
> > > |----------|-----------|-----------|-------------|----------|----------
> > > | WAV db6  | 83.38     | 0.0648    | 21.25    | 31.48 | -1156.63 |
> > > | DnCNN    | 30.21     | 0.0372    | 29.86    | 31.59 | -503.84  |
> > > | DRUNet   | **22.23** | **0.0223**| **32.86**| **34.11**| **-312.66**|
> > >
> > > We observe that the model rankings obtained from both $\Phi_1$ and $\Phi_2$ match those of the Bayesian log evidence, as computed by the Proxnest algorithm. The results are also in agreement with the performance of each model, as reported by PSNR.
> > >
> > > Moreover, for a more rigorous quantitative comparison, we include the analytical calculation of the Bayes factor (BF), i.e. ratio of Bayesian evidences, for the Gaussian example, which is the gold standard for model comparison.
> > > We observe that our proposed model assessment statistic matches the BF when alpha tends to zero, and also that the model with minimum BF matches the model presenting the minimum $\Phi_3$ statistic for different alpha values and dimensions. These results illustrate the agreement of the proposed approach with the golden standard, the Bayes factor, in terms of ranking Bayesian models.
> > >
> > > We hope that the clarifications and additional results provided in this rebuttal address your concerns. If so, we would be grateful if you could consider revisiting your score in light of the revisions.
> > >
> > > References:
> > > - McEwen et al. 2023, Proximal Nested Sampling with Data-Driven Priors for Physical Scientists
> > > - Cai et al. 2022, Proximal nested sampling for high-dimensional Bayesian model selection

---

### Official Review · Reviewer_BCTd · 2026-03-13

**Soundness:** 3
**Presentation:** 3
**Significance:** 3
**Originality:** 3
**Overall Recommendation:** 5
**Confidence:** 3

**Summary:**

This paper focuses on Bayesian model selection and misspecification testing for imaging inverse problems when only a single noisy measurement is available. The main idea is to split one measurement into two pseudo-measurements by noise injection, and then perform a cross-validation-like procedure. The paper introduces a likelihood-based score for likelihood comparison and a posterior-based score for prior assessment. Experiments are conducted on toy Gaussian examples, image deblurring, and MRI reconstruction.

**Compliance With Llm Reviewing Policy:**

Affirmed.

**Final Justification:**

I think my concerns have been addressed. Increase my score accordingly. 4->5

**Key Questions For Authors:**

Please see the weakness section.

**Limitations:**

No limitation is discussed in this paper.

**Strengths And Weaknesses:**

Pro:
- This is an interesting paper. The problem is under-explored and practical.
- The paper is generally clear. The method is easy to follow. The distinction between likelihood-based and posterior-based criteria is also sensible.
- The empirical results are promising. The proposed method performs well in the reported likelihood-selection and OOD-detection experiments.

Con:
- My main concern is about practical cost. The method still requires repeated posterior sampling, sometimes from both split measurements. The paper discusses the trade-off controlled by $\alpha$, but the actual computational overhead is not very clear.
- I also think the method may depend heavily on posterior sample quality. If sampling is biased or not well mixed, the resulting score may also be unreliable. This point is not discussed enough.
- For the posterior-based criterion, the choice of embedding seems important. The paper mentions perceptual embeddings, but it is still unclear how sensitive the results are to that choice in practice.
- The experiments are encouraging, but still somewhat limited relative to the generality of the claims. For example, the MRI OOD setting based on brain versus knee appears fairly separated semantically. More subtle misspecification settings would make the empirical case stronger.

---

> ### Author Rebuttal · Authors · 2026-03-31
>
> We thank the reviewer for their comments and interest in our method.
>
> **Strengths And Weaknesses**
> 1.
> The cost of the proposed method is significantly higher than the cost of computing a point estimator, but otherwise manageable and comparable to the cost of probing other complex aspects of the posterior distribution. This cost depends chiefly on the number measurement splits $K$, and posterior samples $N$ per split. Crucially, the proposed estimators depend on global statistics and hence converge quickly, producing reliable estimates even for small $N$ (see Fig. 19). Also, each split is independent and be run in parallel, and many modern samplers produce excellent samples with a low cost per sample. By comparison, existing unsupervised methods for model selection such as computing the Bayesian model evidence via thermodynamic integration or nested sampling require orders of magnitude more posterior samples and often struggle in high-dimensional inference. We will add a new appendix discussing in detail the computational cost for the proposed approach for each experiment.
>
> 2.
> The reviewer raises an important point. If the sampler performs poorly the results will be unreliable. Having said that, this will manifest clearly in the obtained samples, especially in the context of imaging sciences. Note that alternative Bayesian model comparison and assessment methodology would also exhibit this same fragility to poor sampler performance.
>
> Our experiments show that our method produces reliable model selection and misspecification detection results across different sampler families, suggesting some robustness to moderate sampling performance. We are adding a new experiment where we purposely degrade the sampling quality of the score-based diffusion sampler by reducing the number of discretization steps for the OOD detection experiment.
> We observe that the results remain consistent for a wide, degrading midly with sample quality, until the sampler breaks down (15 steps). This experiment will be included in the revised manuscript.
>
> 3.
> We thank the reviewer for this remark, we agree that the choice of embeddings might have an impact on model selection or OOD detection, as we stated in Section C.1 of the appendix.
>
> In order to assess the sensitivity of the proposed method to a larger choice of embeddings, we carry out an ablation study and compute $\Phi^2_y(\mathcal M)$ with additional embeddings for the OOD experiments, adding the SOTA perceptual embeddings Dreamsim (Fu et al. 2023) and embeddings extracted from the original patch-32 CLIP (Radford, et al. 2021) to the already-tested LPIPS metric and DinoV2 embeddings (Oquab, et al. 2023).
> We report the rejection rates (type 1 error for ID images, power for OOD images) below for $\sigma_\kappa=0.5$:
>
> Embeddings|FFHQ (ID)|Celeb (ID)|Mild OOD (Met)|Strong OOD
> ---|---|---|---|---
> LPIPS|0|6.7|87|100
> DinoV2|0|6.7|87|100
> Dreamsim|13|60|100|100
> CLIP|6.7|0.|6.7|42
>
> CLIP does not produce meaningful results, as the samples produced for this experiment are semantically close. Dreamsim is more discriminative: the reference estimator values on FFHQ are more concentrated for this metric, resulting in a higher rejection rate for ID images. However, this also increases the power for mildly OOD images, even at higher blur levels. We will add the full experiment to the revised article.
>
> 4.
> While we agree with the reviewer that the misspecifications are more obvious in the MRI OOD setting, we would argue that the misspecifications presented in the other experiments are less easily discernible. For instance, a model that was well trained on FFHQ can still produce high quality samples for Met-Faces measurements, as depicted in Fig. 9. Although these samples are far from the ground truth images, identifying these samples as OOD without having access to the ground truth remains a difficult issue. In terms of model selection of section 4.2, the misspecification of the models under scrutiny is very subtle, as it can be seen from the different kernels depicted in Figure 9. It is difficult to visually distinguish models via blurred images for these different kernels, but the proposed method is nevertheless able to reliably select the correct model.
>
> **Additional comments**
>
> We thank the reviewer for their constructive feedback and hope that the additional experiments and clarifications provided in this rebuttal address your concerns. If so, we would be grateful if you could consider revisiting your score in light of the revisions.
>
> References:
> - Fu et al. 2023, DreamSim: Learning New Dimensions of Human Visual Similarity using Synthetic Data\
> - Radford et al. 2021, Learning Transferable Visual Models From Natural Language Supervision\
> - Oquab et al. 2023, Dinov2: Learning robust visual features without supervision

---

> > ### Author Rebuttal · Reviewer_BCTd · 2026-04-04
> >
> > Thanks for the rebuttal. Could the authors provide more numerical results on the first two points?

---

> > > ### Author Response · Authors · 2026-04-08
> > >
> > > We thank the reviewer for their response. Based on their feedback, we have incorporated the two following numerical results:
> > >
> > > 1.
> > > We report below representative run times per image for the experimental setup of Section 4.3.1, *i.e.*, $K=10$ splits and $N=20$ steps, using the DiffPIR sampler originally used in the experiment, as well as a gradient step denoiser prior as in Section 4.2. We draw here a single sample from the posterior distribution $x|y^+$ for each split ($L=1$), resulting in $210$ posterior samples in total.
> > >
> > > Sampler | Total time (s) | Average time per sample (s) |
> > > :--- | :--- | :--- |
> > > PNP (GSD prior)| 323 | 1.54 |
> > > DiffPIR   | 1802 | 8.58 |
> > >
> > > We obtained these results using a single 16GB Nvidia V100. Note that these run times could be greatly improved by making better use of GPU resources (*e.g.*, we only used a batch size of $2$ for the DiffPIR) and by leveraging modern distilled samplers that produce excellent samples in as little as 4-10 NFEs.
> > >
> > > 2.
> > > We add a new experiment where we purposely degrade the sampling quality of a score-based diffusion sampler (DiffPIR) by reducing the number of discretization steps for the OOD detection experiment. We report in the following tables the performance rates in \% (Type 1 error for ID images, Power=1-Type II error for OOD images) for the images of the test dataset, for $\sigma_\kappa=0.5$ and $\sigma_\kappa=2$ respectively. The ID set is composed of the 15 test images from FFHQ and the CelebA images.
> > >
> > > Image subset$\backslash$ N° of DiffPIR diffusion steps | 15 |35 | 75 |150 |300
> > > | :---| :--- | :--- | :--- | :--- |:---
> > > ID|17 | 6.7 | 6.7  |3.3 | 3.3
> > > Mild OOD:  |80 |  80 |  93.|87| 87
> > > Strong OOD|  76 | 82|  98| 100 | 100
> > >
> > > Image subset$\backslash $N° of DiffPIR diffusion steps | 15 |35 | 75 |150 |300
> > > | :---| :--- | :--- | :--- | :--- |:---
> > > ID|3.3 | 10. | 6.7  |10 | 6.7
> > > Mild OOD:  |0.0 |  80 |  87|80| 73
> > > Strong OOD|  0.0 | 96|  100| 100 | 100
> > >
> > > We also computed the supervised PSNR w.r.t. the posterior mean of $x|y^-$, averaged over the $K$ splits for both blur levels:
> > > Image subset$\backslash$ N° of DiffPIR diffusion steps |15 |35|75|150|300
> > > |:-|:-|:-|:-|:-|:-|
> > > ID |27.04| 29.10 | 30.73 |31.4 | 31.36
> > > Mild OOD |26.96| 29.10|  30.91| 31.76| 31.78
> > > Strong OOD |25.94| 27.49| 28.64| 29.03 |28.88
> > >
> > > Image subset$\backslash$ N° of DiffPIR diffusion steps |15|35|75|150|300
> > > |:-|:-|:-|:-|:-|:-|
> > > ID|14.77| 26.59 |29.07 |29.12 |29.01
> > > Mild OOD|14.56| 26.33 |29.98 |30.13| 30.00
> > > Strong OOD|14.22 |22.88 |26.06| 26.15 |26.04
> > >
> > > We observe a reduction in the rejection rate for strongly OOD images as the quality of the generated samples decreases, especially for $\sigma_\kappa=0.5$. This is due to the fact that reconstruction quality remains broadly acceptable for OOD images, making the task of detecting OOD samples harder in the presence of additional reconstruction noise. Moreover, despite the presence of artifacts and overall low sample quality, the estimator still behaves coherently at $35$ iterations for both blur levels. Lastly, the sampler completely breaks down for $\sigma_\kappa=2$ and $15$ iterations, as can be seen in the PSNR table. In this case, our estimators also fail, as they rely on posterior samples. Such catastrophic failure cases would require a different approach.
> > >
> > > We hope that the clarifications and additional results provided in this rebuttal address your concerns. If so, we would be grateful if you could consider revisiting your score in light of the revisions.

---

### Decision · Program_Chairs · 2026-04-30

**Decision:**

Accept (regular)

**Comment:**

This paper proposes an unsupervised method for Bayesian model selection and misspecification testing in imaging inverse problems using data fission with Bayesian cross-validation. The contribution is considered as original.

Scores ranged from 3 to 5. The rebuttal was important, adding comparisons with Proxnest, a degraded-sampler ablation, and additional prior selection experiments. It leads to have the most expert referee to have a positive opinion at the end of the discussion period. Some weaknesses remain, with the main one being that the empirical comparisons were only introduced during the rebuttal and are not yet integrated into the manuscript, so it can be hard to have a final opinion.

 I recommend to accept this contribution to ICML 2026 despite these flaws, but I strongly encourage the authors to polish the final version in the light of the change that was discussed during the rebuttal